

# The sensitivity of Southern Ocean aerosols and cloud microphysics to sea spray and sulfate aerosol production in the HadGEM3-GA7.1 chemistry-climate model

Laura E. Revell[1], Stefanie Kremser[2], Sean Hartery[1], Mike Harvey[3], Jane P. Mulcahy[4], Jonny Williams[3], Olaf Morgenstern[3], Adrian J. McDonald[1], Vidya Varma[3], Leroy Bird[2], and Alex Schuddeboom[1]

[1]School of Physical and Chemical Sciences, University of Canterbury, Christchurch, New Zealand
[2]Bodeker Scientific, Alexandra, New Zealand
[3]National Institute of Water and Atmospheric Research, Wellington, New Zealand
[4]Met Office, Exeter, United Kingdom

**Correspondence:** Laura Revell (laura.revell@canterbury.ac.nz)

**Abstract.** With low concentrations of tropospheric aerosol, the Southern Ocean offers a 'natural laboratory' for studies of aerosol-cloud interactions. Aerosols over the Southern Ocean are produced from biogenic activity in the ocean, which generates sulfate aerosol via dimethylsulfide (DMS) oxidation, and from strong winds and waves that lead to bubble bursting and sea-spray emission. Here we evaluate the representation of Southern Ocean aerosols in the HadGEM3-GA7.1 chemistry-climate model. Compared with aerosol optical depth (AOD) observations from two satellite instruments (the Moderate Resolution Imaging Spectroradiometer, MODIS-Aqua c6.1 and the Multi-angle Imaging Spectroradiometer, MISR), the model simulates too-high AOD during winter and too-low AOD during summer. By switching off DMS emission in the model, we show that sea spray aerosol is the dominant contributor to AOD during winter. In turn, the simulated sea spray aerosol flux depends on near-surface wind speed. By examining MODIS AOD as a function of wind speed from the ERA-Interim reanalysis and comparing it with the model, we show that the sea spray aerosol source function in HadGEM3-GA7.1 overestimates the wind speed dependency. We test a recently-developed sea spray aerosol source function derived from measurements made on a Southern Ocean research voyage in 2018. In this source function the wind speed dependency of the sea spray aerosol flux is less than in the formulation currently implemented in HadGEM3-GA7.1. The new source function leads to good agreement between simulated and observed wintertime AOD over the Southern Ocean, however reveals partially compensating errors in DMS-derived AOD. While previous work has tested assumptions regarding the seawater climatology or sea-air flux of DMS, we test the sensitivity of simulated AOD, cloud condensation nuclei and cloud droplet number concentration to three atmospheric sulfate chemistry schemes. The first scheme adds DMS oxidation by halogens and the other two test a recently-developed sulfate chemistry scheme for the marine troposphere; one tests gas-phase chemistry only while the second adds extra aqueous-phase sulfate reactions. We show how simulated sulfur dioxide and sulfuric acid profiles over the Southern Ocean change as a result, and how the number concentration and particle size of the soluble Aitken, accumulation and coarse aerosol modes are affected. The new DMS chemistry scheme leads to a 20% increase in the number concentration of cloud condensation





nuclei and cloud droplets, which improves agreement with observations. Our results highlight the importance of atmospheric chemistry for simulating aerosols and clouds accurately over the Southern Ocean.

# 1 Introduction

Clouds and aerosols play an important role in Earth's energy balance by absorbing and scattering solar and terrestrial radiation.
However, aerosol-radiation and aerosol-cloud interactions are leading sources of uncertainty in determining human influences on climate (Myhre and Shindell, 2013). The Southern Ocean, one of the cloudiest regions on Earth, is remote from anthropogenic sources of aerosol thus making it an ideal environment in which to study aerosol-cloud interactions (Hamilton et al., 2014). Clouds forming in pristine regions such as over the Southern Ocean are highly sensitive to aerosol perturbations (Koren et al., 2014), however the specific roles that marine aerosols play in cloud formation are highly uncertain (Brooks and Thornton, 2018).

Marine aerosols are either primary or secondary in origin. Primary aerosols such as sea spray are directly injected into the atmosphere when breaking waves entrain air bubbles, which subsequently form whitecaps and burst. Secondary aerosols such as sulfate aerosol are formed from nucleation of sulfur-containing gases. Sea spray aerosol (SSA) is generated in significant quantities over the Southern Ocean by strong winds and waves (Murphy et al., 1998). SSA is an important contributor to the global-mean clear-sky AOD (Shindell et al., 2013), and its production is highly dependent on wind speed (Smirnov et al., 2003; Mulcahy et al., 2008; Glantz et al., 2009). A significant component of primary marine aerosol is sea salt with some fraction of organics (Fossum et al., 2018). Marine organic aerosols, along with sulfate aerosols, result from biogenic activity in the ocean (O'Dowd et al., 2004). Marine phytoplankton produce dimethylsulfoniopropionate (DMSP), which is broken down into several products including dimethylsulfide (DMS). Oceanic DMS emissions are the main source of atmospheric sulfur over the Southern Ocean, with an estimated 28.1 TgS transferred from the oceans globally into the atmosphere each year (Lana et al., 2011). Around coastal Antarctica, melting of sea ice elevates the seawater DMS concentration (Trevena and Jones, 2006), leading to a seasonal anti-correlation between sea ice extent and aerosol concentration (Gabric et al., 2018). When DMS is emitted into the atmosphere, it has a lifetime of 1–2 days and undergoes a series of chemical reactions to form sulfur dioxide ($SO_2$) which is further oxidised to form sulfate aerosol.

Aerosol particles emitted into the atmosphere can grow in size via condensation and coagulation. Depending on the aerosol composition and meteorological conditions such as the cloud base updraft velocity (Rosenfeld et al., 2014), particles larger than $\gtrsim 50$ nm in diameter can be "activated" to cloud condensation nuclei (CCN) around which water vapour can condense and cloud droplets form. Generally speaking, liquid water clouds which have been perturbed by aerosols consist of more but smaller cloud droplets, and therefore scatter radiation more efficiently (Twomey, 1977; Boucher and Randall, 2013).

Previous work has confirmed that cloud droplet number concentrations ($N_d$) over the Southern Ocean are correlated with marine biogenic activity (Thomas et al., 2010; Woodhouse et al., 2010). Meskhidze and Nenes (2006) identified that observed $N_d$ over a large phytoplankton bloom was twice as large compared to a region distant from the bloom. More recently, McCoy et al. (2015) found that $N_d$ is spatially correlated with regions of high chlorophyll-$a$, and that the spatiotemporal variability in



$N_d$ is found to be driven mostly by high concentrations of sulfate aerosol at lower southern latitudes and by organic matter in sea-spray aerosol at higher latitudes.

The models participating in the fifth phase of the Coupled Model Intercomparison Project (CMIP5) simulated Southern Ocean sea surface temperature (SST) biases which are primarily linked to cloud-related errors in shortwave radiation (Hyder et al., 2018). SST biases affect the position of the storm track (Ceppi et al., 2014), which leads to cascading errors in global climate models across the Southern Hemisphere and reduces confidence in projections of climate change and climate extremes in this region (Trenberth and Fasullo, 2010).

To understand potential connections between the representation of aerosols and clouds via the aerosol indirect effect, we investigate the representation of marine aerosols over the Southern Ocean in the Hadley Centre Global Environmental Model version 3, Global Atmosphere 7.1 (HadGEM3-GA7.1). An evaluation of cloud representation in the predecessor model HadGEM3-GA7.0 suggests that significant errors exist in the cloud scheme over the Southern Ocean, but they partially compensate one another (Schuddeboom et al., 2019). Furthermore, the aerosol forcing and climate feedback in this model is highly sensitive to the representation of DMS-derived sulfate aerosol (Bodas-Salcedo et al., 2019).

HadGEM3-GA7.1 is described in Section 2.1, and simulated AOD is evaluated relative to observations in Section 3.1. We then show how biases in simulated AOD during winter months can be addressed by implementing a new SSA source function derived from measurements collected on the Southern Ocean (Section 3.2). Finally, while much prior work has focussed on testing the sensitivity of Southern Ocean clouds and aerosols to the choice of DMS seawater climatology and/or the DMS sea-air transfer function (Mahajan et al., 2015; Boucher et al., 2003; Fiddes et al., 2018; Korhonen et al., 2008; Woodhouse et al., 2010), we have investigated atmospheric DMS chemistry. We performed sensitivity tests in which different gas-phase and aqueous-phase sulfate chemistry schemes have been implemented. The resulting changes in simulated aerosols and cloud microphysics are shown in Section 3.3.

## 2 Methods

### 2.1 Model description

Simulations were performed with the Hadley Centre Global Environmental Model version 3, Global Atmosphere 7.1 (HadGEM3-GA7.1) (Walters et al., 2019; Mulcahy et al., 2018), which exhibits more realistic aerosol effective radiative forcing compared with preceding versions (Mulcahy et al., 2018). Aerosol emission, evolution and deposition are simulated with the Global Model of Aerosol Processes (GLOMAP-mode), in which sulfate, sea-salt, black carbon and particulate organic matter aerosol are represented in five log-normal size modes. These correspond to particle size ranges of $\leq 10$ nm (nucleation mode), 10–100 nm (Aitken mode), 100–1000 nm (accumulation mode) and $\geq 1000$ nm (coarse mode) (Mann et al., 2010). All modes are soluble, and an insoluble Aitken mode is also included. Mineral dust is represented in the model using a bin emission scheme (Woodward, 2001).

Aerosol-cloud interactions are represented via the UKCA-Activate scheme (West et al., 2014), which simulates the number of aerosols activated into cloud droplets. CCN are defined as aerosols with a diameter $\geq 50$ nm, which is the minimum size





of aerosol that activates with a supersaturation of approximately 0.3% (Lee et al., 2013). The number of activated aerosols is calculated via Köhler theory and depends on aerosol size, composition and number, along with the local temperature, pressure and vertical velocity (Abdul-Razzak and Ghan, 2000). Because the grid cell sizes in global models are too large to resolve cloud base updraft velocity, a probability density function represents the likely distribution of vertical velocity within each grid-box at each time step. The cloud droplet number concentration ($N_d$) is calculated from the number of activated aerosols at the cloud base, weighted by this probability density function (Mulcahy et al., 2018). The number of cloud droplets subsequently influences the cloud albedo, as clouds with larger $N_d$ (and smaller droplets) are optically brighter (Twomey, 1977).

HadGEM3-GA7.1 scales marine DMS emissions by a factor of 1.7 to account for missing sources of marine organics, which yields a better representation of $N_d$ compared with observations (Mulcahy et al., 2018). Here we use a modified configuration of the model, GA7.1-mod, which includes marine organics instead of DMS emission scaling. Furthermore, the GA7.1 standard configuration uses a simplified chemistry scheme, whereby chemical oxidants such as $O_3$, OH, $NO_3$ and $HO_2$ are prescribed as "offline" monthly-mean climatologies in order to reduce computational time. In this study, the model used an online chemistry scheme, StratTrop (also known as CheST – Chemistry of the Stratosphere and Troposphere), which is a combination of the stratospheric and tropospheric chemistry schemes described by Morgenstern et al. (2009) and O'Connor et al. (2014), respectively.

The StratTrop scheme uses a Newton-Raphson solver, and accounts for DMS oxidation via the gas-phase and aqueous-phase reactions shown in Table 1. The oxidation of DMS by OH proceeds by both an addition and abstraction pathway (the first two reactions listed in Table 1), and can produce $SO_2$ and methane sulfonic acid (MSA). The relative yields of these products are important as $SO_2$ leads to new particle formation, while other products such as MSA condense on existing particles therefore increasing their size (von Glasow and Crutzen, 2004; Hoffmann et al., 2016).

Gas-phase $SO_2$ enters the liquid phase via an equilibrium approach (Warneck, 2000) described by Henry's Law. Because $SO_2$ dissociates in the aqueous-phase (Reactions R1 and R2), it is more soluble than the equilibrium Henry's law constant ($K_H$) implies.

$$SO_2 \rightleftharpoons H^+ + HSO_3^- \tag{R1}$$

$$HSO_3^- \rightleftharpoons H^+ + SO_3^{2-} \tag{R2}$$

Therefore, the model uses an effective constant ($K_{H_{eff}}$) which for $SO_2$ is related to $K_H$ by Eq. (1).

$$K_{H_{eff}} = K_H(1 + \frac{k_{R1}}{[H^+]} + \frac{k_{R1}k_{R2}}{[H^+]^2}) \tag{1}$$

$k_{R1}$ and $k_{R2}$ are the equilibrium constants for the aqueous-phase dissociations shown in Reactions R1 and R2. The hydrogen ion concentration ($H^+$) is set as a global number in the model, equivalent to a constant pH of 5.





SSA is generated via a wind speed-dependent parametrisation based on whitecap coverage (Gong, 2003). This function is based on the semi-empirical function by Monahan et al. (1986), but improves the representation of small particles less than 0.1 $\mu$m in diameter. According to Gong (2003), the number of seawater droplets generated per square-meter of sea surface, per increment of particle radius over 20 size bins is calculated via Eq. (2):

$$\frac{dF}{dr} = 1.373u_{10}^{3.41}r^{-A}(1 + 0.057r^{3.45}) \times 10^{1.607e^{-B^2}}$$ (2)

The exponential terms $A$ and $B$ are defined by Eq. (3) and (4):

$$A = 4.7(1 + \Theta r)^{-0.017r^{-1.44}}$$ (3)

$$B = \frac{(0.433 - log(r))}{0.433}$$ (4)

Where $r$ is the particle radius at a relative humidity of 80%, $\Theta$ is an adjustable parameter that controls the shape of the size distributions and $u_{10}$ is the scalar horizontal wind-speed at 10 m above the surface.

## 2.2 Simulations performed

A 20 year reference simulation ("REF") was performed from 1989-2008 to evaluate the model. SSTs and greenhouse gas concentrations were based on observations, and emissions of aerosols and their precursors were prescribed based on the year 2000 (Lamarque et al., 2010). A further eight sensitivity simulations were performed, each of 10 years' duration from 1989–1998. These were designed to test the sensitivity of simulated aerosols to the choice of SSA source function and sulfate chemistry scheme, and are summarised in Table 2. All simulations used the DMS seawater climatology of Lana et al. (2011) and the DMS sea-air exchange parametrisation of Liss and Merlivat (1986). Simulations were run with N96 horizontal resolution (i.e. grid sizes $1.875° \times 1.25°$ in size) and 85 levels between the surface and 85 km.

Analysis of aerosol measurements made on a 2018 *Tangaroa* research voyage on the Southern Ocean indicate that the dependency of SSA production on near-surface wind speed ($u_{10}^{3.41}$) is overestimated by a factor of 2–4 via the Gong (2003) source function (Eq. (2)). Ongoing research by Hartery et al. indicates that Eq. (5) with SSA production dependent on $u_{10}^{2.8}$ is a better fit to observed SSA concentrations in an environment dominated by high wind speeds such as the Southern Ocean. The "SSF" (SSA Source Function) simulation therefore aims to test this using HadGEM3-GA7.1-mod. CHEM1-SSF, CHEM2-SSF and CHEM3-SSF also use the SSA source function described by Eq. (5), in combination with different sulfate chemistry schemes as described below.

$$\frac{dF}{dr} = 2.6u_{10}^{2.8}r^{-A}(1 + 0.057r^{3.45}) \times 10^{1.607e^{-B^2}}$$ (5)

DMS oxidation chemistry is complex (von Glasow and Crutzen, 2004), however the set of reactions describing the conversion of gaseous DMS into sulfate aerosol in StratTrop (Table 1) is simplified due to the computational cost of calculating





chemical reaction rates. We tested three alternative reaction schemes with incremental increases in complexity, with the aim of identifying how sensitive Southern Ocean aerosols and clouds are to the choice of chemistry scheme. The three sulfate chemistry schemes investigated in our CHEM1, CHEM2 and CHEM3 simulations are described in Table 3. The CHEM1 and CHEM2 sensitivity simulations use the same aqueous-phase sulfate chemistry scheme as REF (i.e. the default StratTrop scheme included in HadGEM3-GA7.1-mod), but with increased complexity of the gas-phase chemistry. CHEM1 includes DMS oxidation by halogens as they have been shown to play an important role in the remote marine atmosphere (Boucher et al., 2003; von Glasow and Crutzen, 2004; Chen et al., 2018). CHEM2 includes a gas-phase DMS oxidation scheme based on the scheme recently developed for the marine troposphere by Chen et al. (2018). CHEM3 is identical to CHEM2 except that additional aqueous-phase sulfate reactions are included. This scheme is based on the aqueous-phase scheme by Chen et al. (2018), however the oxidation reactions by OH are excluded as OH uptake into cloud droplets is subject to numerous uncertainties (Chen et al., 2018) and is not currently implemented in HadGEM3-GA7.1-mod. The new scheme also includes aqueous-phase treatment of MSA, which is treated as a sink of DMS in HadGEM3-GA7.1-mod and does not oxidise to form aerosol. In the NODMS simulation, DMS emissions are switched off to help isolate its role in the annual AOD cycle over the Southern Ocean.

## 2.3 Observational data sets

### 2.3.1 Satellite-based observations

Model output is compared to daily-mean aerosol optical depth (AOD) data derived from Moderate Resolution Imaging Spectroradiometer (MODIS)-Aqua measurements, collection 6.1 (Platnick et al., 2003; Sayer et al., 2014) and monthly-mean AOD derived from the Multi-angle Imaging Spectroradiometer (MISR). MODIS is a passive imaging radiometer that measures reflected solar and emitted thermal radiation across a 2330 km swath, providing near-daily global coverage over land and ocean at the Equator and overlap between orbits at higher latitudes. MODIS was deployed on the Aqua satellite in May 2002. Here the MODIS Level 3 data product with a spatial resolution of $1° \times 1°$ (latitude/longitude grid) is used for AOD at 550 nm. A number of inconsistencies and potential retrieval problems, which have been identified in past MODIS products, have been remedied in MODIS collection 6.1. The data used in this study were obtained using the combined Deep Blue (land retrieval only) and Dark Target (ocean and land retrieval) approaches (Sayer et al., 2014). In this study we use MODIS measurements from 2003 to 2007; a period characterised by a notable absence of volcanic eruptions reaching the lower stratosphere as discussed below. Since MODIS data are limited at high latitudes in the visible band, we spatially and temporally co-locate MODIS and model data before calculating climatological monthly means (Schutgens et al., 2016).

A previous study by Remer et al. (2008) showed that over oceans, MODIS-retrieved AOD agrees well (within the expected uncertainties) with observations obtained from the ground-based Aerosol Robotic Network of sun photometers (AERONET, Holben et al., 1998) more than 60% of the time. Toth et al. (2013) subsequently showed that collection 5 MODIS data overestimates AOD as observed from AERONET sites at mid-to-high southern latitudes (Dunedin (45.8°S, 170.5°E) and Crozet Island (46.4°S, 51.9°E)), and this bias is attributed to cloud contamination of the MODIS AOD product. Since then, MODIS



data have been reprocessed, implementing a number of improvements in the retrieval algorithm, including the use of a revised cloud-mask to account for cloud contamination.

MISR was deployed on Terra, NASA's first Earth Observing System (EOS) spacecraft, on December 18, 1999. MISR views the sunlit Earth simultaneously at nine widely-spaced angles in four visible and near infrared wavelengths, with a swath of

approximately 400 km (Diner et al., 1998). Due to the overlap of the swathes near the poles and their wide separations at the equator, coverage time varies from 2 to 9 days, respectively. The MISR AOD product has been validated with respect to AERONET (Kahn et al., 2010; Garay et al., 2017), and also shows good agreement with AOD Level 3 data from MODIS (Mehta et al., 2016).

Here the MISR Level 3 data product with a spatial resolution of $0.5° \times 0.5°$ is used for total AOD at 555 nm, with the

AOD retrieval algorithm dependent on surface types such as vegetated areas, dark water bodies and high contrast terrain (Martonchik et al., 2009). To match the MODIS observation period, MISR measurements from 2003–2007 are used in this study. The measurements are temporally co-located with the model data before calculating climatological monthly means.

AOD quantifies the amount of aerosol in the vertical column between the Earth's surface and the top of the atmosphere. Owing to the lack of large volcanic eruptions during the period of study, the stratospheric component of AOD over the Southern

Ocean is around 0.007 (an absolute value) as indicated by the global stratospheric aerosol data set compiled by Thomason et al. (2018). Therefore, tropospheric aerosols are the dominant contributor to total AOD over the Southern Ocean between 2003–2007. Satellite-derived AOD depends on the atmospheric concentration of particulate matter such as sea spray, mineral dust, organic compounds and sulfate originating from the oxidation of atmospheric DMS. As the Southern Ocean is remote from anthropogenic influence, the predominant tropospheric aerosols contributing to AOD are sulfate and sea spray (Gabric et al.,

20  2005).

### 2.3.2   In-situ and flask measurements of DMS

Surface observations of DMS, which is an important precursor of sulfate aerosol, are relatively rare in the Southern Ocean and Antarctic region. In this study, we rely on observations from research voyages and the measurement stations at Amsterdam Island in the southern Indian Ocean (38°S, 78°E) and Cape Grim, Tasmania (41°S, 145°E). Atmospheric flask measurements

were obtained from Amsterdam Island between 1987–2008 (Sciare et al., 2001) and Cape Grim between 1989–1996 (Ayers and Gillett, 2000). Ship-borne measurements were obtained from the SOAP (Surface Ocean Aerosol Production) campaign, which sampled between 42-47°S, 172-180°E during February and March 2012 (Law et al., 2017; Bell et al., 2015; Smith et al., 2018), and SOIREE (the Southern Ocean Iron RElease Experiment), which sampled between 42-63°S, 139-172°E in February 1999 (Boyd and Law, 2001; Boyd et al., 2000). During SOIREE, DMS was measured in discrete water samples from vertical

profiles and whilst underway in air and surface water (Turner et al., 2004). During SOAP, surface water and surface microlayer DMS were measured (Walker et al., 2016) in addition to atmospheric DMS concentrations and emission fluxes (Bell et al., 2015; Smith et al., 2018).



## 3 Results and Discussion

### 3.1 Evaluation of simulated aerosol optical depth (AOD)

Figure 1 shows climatological monthly zonal-mean AOD between 35–65°S as simulated by HadGEM3-GA7.1-mod and observed by MODIS and MISR. The seasonality in AOD over the Southern Ocean is similar between MODIS and MISR, as

shown previously by Ocko and Ginoux (2017). The model generally agrees with the maximum, minimum and mean AOD observed by MODIS (Fig. 1a,b). However, the simulated seasonal cycle is out-of-phase. The model simulates too much aerosol in winter (JJA) and too little in summer (DJF) compared with satellite observations (Fig. 1c,d).

As discussed earlier, sulfate aerosol from biogenic sources and SSA predominantly contribute to AOD over the Southern Ocean. By performing a simulation with DMS emissions switched off (the NODMS simulation) and comparing it with the

REF simulation (Fig. 2d), it is apparent that the model simulates primarily SSA during winter (July and August, 50–65°S). This result is consistent with the Aerocom (Aerosol Comparisons between Models and Observations) phase II models, which simulate a seasonal maximum in sea salt AOD during winter at southern high latitudes, while sulfate AOD maximises in summertime.

In DJF AOD is approximately 60% lower in the NODMS simulation compared with REF, indicating that sulfate aerosol of

marine biogenic origin is primarily produced during summertime when increased solar radiation and warmer waters make the ocean more biologically productive. Indeed, measurements at Baring Head (41°S, 179°E) indicate that sulfate in fine aerosol modes is mostly secondary sulfate from marine DMS emission, exhibiting an annual maximum in summertime, while coarse sulfate aerosol is mainly from sea salt, and is relatively constant throughout the year (Li et al., 2018).

Total AOD is calculated in HadGEM3-GA7.1 from adding together the individual contributions of dust AOD and the Aitken

mode (soluble + insoluble), accumulation mode and coarse mode AODs. Aerosol particles in the soluble modes may activate to cloud condensation nuclei, and the contribution to total AOD from these modes is shown in Fig. 2a–c. Coarse mode aerosol is the major contributor to total AOD due to its size, and maximises in Southern Hemisphere autumn, winter and spring (Fig. 2c), implying that it is mostly SSA as discussed above. Accumulation mode aerosol (Fig. 2b) shows a clear seasonal cycle which maximises during summer (as shown previously by e.g. Vallina et al. (2007)), indicating that this is mostly aerosol of marine

biogenic origin. Aitken and accumulation mode aerosol increases during springtime between ∼35–40°S, associated with long-range transport of particulate matter from South America, Australia and South Africa (McCluskey et al., 2019).

### 3.2 Simulated sea salt aerosol

Given that HadGEM3-GA7.1-mod primarily simulates SSA during winter, we now examine SSA production in more detail. Zonal-mean near-surface wind speeds between 40–60°S are shown in Fig. 3a. The model agrees reasonably well with the ERA-

Interim reanalysis (Dee et al., 2011), at least in the zonal mean. However, the actual position of the storm track tends to be zonally shifted in the model (not shown) which is associated with the model's shortwave radiation bias discussed in Section 1. While sparse observations over the Southern Ocean lead to some uncertainty regarding the comparative accuracy of near-surface wind data sets in reanalyses, Bracegirdle et al. (2013) indicate that ERA-Interim is the most reliable of contemporary





reanalyses. Furthermore, ERA-Interim near-surface winds agree well with other reanalyses such as the National Centers for Environmental Protection Climate Forecast System Reanalysis (NCEP/CFSR) and NASA Modern Era Retrospective-Analysis for Research and Applications (MERRA) (Bracegirdle et al., 2013).

Wind speeds over the Southern Ocean show a clear seasonality maximising between autumn and spring, and thus we expect
more SSA to be produced during this time. As described in Section 2.1, the model uses the SSA source function of Gong (2003) in which SSA generation scales according to a power law with wind speed. The correlation between simulated AOD and wind speed is shown in Fig. 4a. Comparing this with a similar regression model fit derived from MODIS and ERA-Interim data (Fig. 4b) illustrates that the wind speed dependency of SSA production over the Southern Ocean is overestimated in the model as evidenced by the regression model fits obtained. This is supported by SSA measurements made on the 2018 *Tangaroa*
research voyage, which indicate that SSA production requires a threshold wind speed, below which no SSA is produced. The measurements also show that the SSA flux predicted by the Gong (2003) parametrisation increases too quickly as a function of wind speed. These two effects result in over-production of SSA at all wind speeds by a factor of 2–4 (Hartery et al.).

AOD over the Southern Ocean in the SSF simulation using the Hartery et al. source function (see Eq. (5) and Table 2) is shown in Fig. 3b. Compared with the REF simulation (Fig. 3c) the reduction in AOD is reasonably uniform throughout
the year, with the reduction in coarse mode AOD (shown for REF in Fig. 2c) between March and November particularly visible. Comparing to MODIS observations, the Hartery et al. source function performs well during winter months when SSA is the dominant contributor to AOD (Fig. 3d). Changes in aerosol mode number concentrations and dry diameters in the SSF simulation are discussed later in Section 3.3.

Our finding that the SSA contribution to AOD is overestimated in the REF simulation is consistent with the Atmospheric
Chemistry and Climate Model Intercomparison Project (ACCMIP) models, which overestimate annual-mean sea-salt AOD between 50–60°S compared to observations from AERONET sun photometers (Shindell et al., 2013). Our results are also consistent with previous work suggesting that the Gong (2003) source function overestimates the SSA dependency on wind speed (Madry et al., 2011). Jaeglé et al. (2011) found that using the GEOS-Chem chemical transport model, the Gong (2003) SSA source function led to overestimation of coarse mode SSA in the atmosphere by a factor of two-to-three at high wind speeds,
and suggested that the discrepancies are dependent on sea-surface temperature. Similarly, Spada et al. (2015) showed that sea-salt surface concentrations are overestimated compared with observations at southern high latitudes in chemical transport model simulations using the Gong (2003) source function. However, by implementing a weighting factor based on sea-surface temperature as suggested by Jaeglé et al. (2011), their model simulated SSA concentrations that are in closer agreement with observations (Spada et al., 2015).

How SSA should ultimately be represented in global models remains the subject of ongoing research. Along with Jaeglé et al. (2011), other studies have found that SSA concentrations are correlated with sea surface temperature (Mårtensson et al., 2003; Sellegri et al., 2006; Sofiev et al., 2011; Grythe et al., 2014). More recently, Forestieri et al. (2018) demonstrated that variability in seawater composition may have just as large an impact on SSA production as temperature. Nonetheless, our results demonstrate that for the Southern Ocean during winter when SSA is the dominant contributor to AOD, reducing the wind speed
dependency of SSA production results in good agreement between the model and observations. However, Fig. 3d points to the





existence of partially compensating errors, namely that sulfate aerosol is underestimated in the model during summertime even more than suggested by the REF simulation. Given the importance of sulfur chemistry in the marine atmosphere, we now discuss the CHEM simulations and sensitivity of simulated aerosols and cloud microphysics to the choice of sulfate chemistry scheme.

## 3.3 DMS oxidation chemistry

Seawater DMS from the Lana et al. (2011) climatology used as input to HadGEM3-GA7.1-mod is shown in Fig. 5. Seawater DMS concentrations maximise in austral summer along the Antarctic continent following sea ice melt, and the corresponding release of aerosol precursors by phytoplankton which grow on the underside of sea ice (Gabric et al., 2005). In the Lana et al. (2011) climatology the maximum summertime DMS concentration reached at southern high latitudes during DJF is up to 15 nM lower than in the older Kettle and Andreae (2000) seawater DMS climatology. Using the ECHAM5-HAMMOZ model, Mahajan et al. (2015) showed that use of the Lana et al. (2011) climatology improved the simulation of DMS at Amsterdam Island, particularly during summertime when observed concentrations are large. However, the Lana et al. (2011) climatology includes large uncertainties as it was compiled from cruise observations interpolated to make a global climatology. These uncertainties translate to variations in $N_d$ between 2–5 $cm^{-3}$ in HadGEM3-GA7.1 (Mulcahy et al., 2018).

Simulated surface atmospheric DMS concentrations in the REF simulation agree reasonably well with measurements from the SOAP and SOIREE voyages (Fig. 6a), although the spread in measurements varies by almost 1000 ppt. The model does not capture such a large spread in variability; likely because it provides output averaged over coarse horizontal grid cells and SOAP sought out the highest chlorophyll/DMS-containing waters at the time of the voyage. To examine the seasonal cycle in atmospheric DMS, we compare model results with measurements obtained from Amsterdam Island (Fig. 6b) and Cape Grim (Fig. 6c). For both observations and the model, the summertime maximum coincides with the peak of phytoplankton productivity. At Amsterdam Island the REF simulation underestimates DMS in January by 55% and overestimates it in July by a factor of three. However at Cape Grim, DMS is overestimated throughout the year in the REF simulation, and simulates approximately five times too much DMS in January. The large DMS concentrations simulated at Cape Grim relate to the Lana et al. (2011) seawater DMS climatology, which shows a region of high DMS productivity close by (Fig. 5b).

In all three CHEM simulations, the change in the simulated surface atmospheric DMS concentration is negligible relative to the magnitude of the seasonal cycle in DMS (Fig. 6b,c). Fig 7a shows how DMS changes through the lowest 2 km of the atmosphere over the Southern Ocean. In the CHEM simulations the DMS concentration is 7–13% larger than in REF. This likely relates to the rate constant for the first DMS + OH reaction listed in Tables 1 and 3, which is an order of magnitude smaller in the new scheme tested compared with the existing StratTrop scheme, implying that it will proceed more slowly and therefore less DMS will be oxidised.

$SO_2$ concentrations decrease with height above 0.5 km altitude (Fig. 7b), which is the approximate cloud base (Fig. 9). Surface $SO_2$ concentrations are almost 30 ppt lower in the CHEM1 simulation compared with REF. This is likely due to the implementation of reactions between DMS and halogens (DMS+BrO and DMS+Cl), which may convert the sulfur in DMS to DMSO and $SO_2$ (rather than only $SO_2$; see Table 3). In particular, the DMS+BrO reaction has been shown to be





particularly important in the remote marine troposphere (Chen et al., 2018; Boucher et al., 2003; von Glasow and Crutzen, 2004). Measurements at Baring Head (41°S, 179°E) during February and March 2000 indicate that the $SO_2$/DMS ratio of clean boundary layer air originating from over the Southern Ocean is approximately 0.06–0.26 (de Bruyn et al., 2002). Our simulated ratios of surface $SO_2$/DMS over the Southern Ocean (40–60°S) agree with this measured range for the REF, CHEM1,

CHEM2 and CHEM3 experiments (0.19, 0.11, 0.15 and 0.13, respectively).

    Examining the global $H_2SO_4$ distribution in the REF simulation reveals that $H_2SO_4$ mixing ratios over the Southern Ocean in DJF are larger than any other region (not shown), consistent with the ECHAM-HAMMOZ model (Thomas et al., 2010). $H_2SO_4$ concentrations are increased by $\sim$0.015–0.025 ppt relative to REF in the CHEM2 and CHEM3 simulations (Fig. 7c) due to the extra DMS oxidation reactions added.

Figure 8 shows vertical profiles of aerosol mode number concentration and particle dry diameter over the Southern Ocean in the REF and sensitivity simulations. For reference, the mean mass fraction of cloud liquid water in DJF over the Southern Ocean is shown in Fig. 9 to illustrate that the aerosol profiles we examine are situated within the cloud layer. In the SSF simulation, decreasing the dependency of SSA generation on wind speed means that the number concentration of accumulation and coarse mode particles decreases by 30–50% (Fig. 8b and c). The particle dry diameters in these modes are largely unchanged (Fig. 8e

and f). However, the soluble Aitken mode changes; the number concentration increases by $\sim$40% in the SSF simulation relative to REF and the average particle dry diameter decreases by 10 nm (Fig. 8a and d). Initially this was unexpected, as SSA is emitted only into the accumulation and coarse modes in the model, and not the Aitken mode (Mann et al., 2010). The change in the Aitken mode likely comes from smaller non-SSA particles (e.g. sulfate aerosol) being unable to coagulate on larger SSA particles as these are reduced in number.

In the CHEM simulations, the coarse mode remains largely unchanged regardless of the chemistry scheme used (Fig. 8c,f). In CHEM2 and CHEM3 simulations there are more smaller particles in the accumulation mode which are smaller on average (Fig. 8b,e), which has implications for cloud microphysics. As discussed earlier, soluble aerosols such as sea salt and sulfate with a diameter $\geq$50 nm can become activated to CCN, corresponding to a supersaturation of $\sim$0.3%. Simulated CCN and $N_d$ over the Southern Ocean are shown in Fig. 10. In the REF simulation, summertime-mean CCN concentrations at 800

m above the surface average 120 cm$^{-3}$, which is the same as measurements reported at Cape Grim (41°S, 145°E, 0.23% supersaturation, Korhonen et al., 2008) and Princess Elisabeth Antarctic Research Station (72°S, 23°E, 0.3% supersaturation, Herenz et al., 2019).

    At southern high latitudes, the number fraction of SSA CCN is larger than any other region on the globe (Quinn et al., 2017). Therefore owing to the reduced aerosol abundance in the SSF simulation, CCN concentrations also decrease by up to $\sim$13%

relative to REF (Fig. 10a). In the CHEM simulations, CCN concentrations decrease by -18% (CHEM1) to +25% (CHEM2 and CHEM3), which is likely linked to the changes in accumulation mode aerosol shown in Fig. 8. The changes in CCN in the CHEM simulations translate to changes in $N_d$ over the Southern Ocean of -13% in CHEM1 to +20% in CHEM2 and CHEM3 (Fig. 10b). Bodas-Salcedo et al. (2019) show that in HadGEM3-GA7.1, the simulated seasonal cycle in $N_d$ over the Southern Ocean is primarily driven by seawater DMS emissions. While the model captures the observed seasonality in $N_d$, the





magnitude is too low, which was also reported by Mulcahy et al. (2018). However, the CHEM2 and CHEM3 simulations bring the model into better agreement with $N_d$ observations.

Of all the CHEM and CHEM-SSF sensitivity simulations, AOD simulated in the CHEM1 simulation agrees most favorably with MODIS (Fig. 11a), and the root-mean square error of 0.029 is the same as it is when comparing REF and MODIS (Fig. 1c). However, the seasonal bias remains. The CHEM1-SSF simulation shows good agreement with MODIS during austral winter but underestimates summertime AOD and $N_d$ (Fig. 10 and 11d). CHEM2-SSF and CHEM3-SSF show the reverse; simulated summertime AOD agrees well with MODIS but wintertime AOD is too high, even with the new SSA source function included. However, given that the chemistry schemes used in the CHEM2 and CHEM3 simulations also show the best agreement with $N_d$ observations, we recommend a combination of the Hartery et al. SSA source function and either the CHEM2 or CHEM3 DMS chemistry schemes for future studies.

A large source of uncertainty in our investigation into aqueous-phase chemistry lies with the constant cloud water pH in the model (assumed to be 5 everywhere). Changes in cloud water pH have substantial impacts on aerosol particle size distributions and CCN concentrations, particularly in parts of the Northern Hemisphere where reductions in anthropogenic $SO_2$ emissions since the mid-1980s have increased cloud water pH (Schwab et al., 2016). Turnock et al. (2019) show that the effect of pH on particles larger than 50 nm in diameter (which may activate to CCN) over the Southern Ocean is not negligible. Aqueous-phase chemistry may also be affected in the model due to the lack of persistent low-lying cloud over the Southern Ocean (Kuma et al., 2019). Aqueous-phase chemistry is more efficient at processing sulfur-containing gases than gas-phase chemistry, but cloud droplets are needed to allow in-cloud droplet chemistry to occur. Future work will focus on these issues, and on evaluating changes to clouds and aerosols outside the Southern Ocean region when these changes are implemented.

## 4 Conclusions

AOD over the Southern Ocean in the HadGEM3-GA7.1-mod CCM exhibits seasonal biases compared with MODIS-Aqua collection 6.1 and MISR satellite observations. The model produces too much aerosol in winter (JJA), and too little in summer (DJF). Simulated AOD in winter consists almost entirely of SSA, the production of which depends heavily on the near-surface wind speed. A comparison of MODIS-observed AOD and ERA-Interim wind speeds indicates that the existing SSA source function in the model overestimates the SSA-wind speed dependence. We tested a new SSA source function in which the wind speed dependency is reduced to match SSA measurements made on the 2018 *Tangaroa* research voyage on the Southern Ocean. Simulated wintertime AOD agrees favourably with observations as a result, but points to partially compensating errors in the formulation of sulfate aerosol production, which maximises over the Southern Ocean in summer as a result of marine biogenic activity. We performed simulations to test the sensitivity of Southern Ocean clouds and aerosols to alternative gas-phase and aqueous-phase chemistry schemes associated with sulfate aerosol. The schemes tested here lead to changes in simulated DMS, $SO_2$, $H_2SO_4$ and aerosol particle sizes and number concentrations. In particular, the CHEM2 and CHEM3 schemes tested lead to increases in CCN and $N_d$ of up to 20%, leading to better agreement between simulated and observed $N_d$. We recommend a combination of the Hartery et al. SSA source function and either the CHEM2 or CHEM3 DMS chemistry

schemes for future studies focussed on the Southern Ocean. Our results underscore the importance of atmospheric chemistry for simulating aerosols and cloud microphysics accurately, and imply that future changes in wind speeds or atmospheric composition associated with anthropogenic climate change may impact cloud and aerosol formation over the Southern Ocean, with implications for the radiative balance in this region.

*Data availability.* MODIS and MISR observations were accessed via the Giovanni online data system, developed and maintained by the NASA GES DISC https://giovanni.gsfc.nasa.gov/ (last access: 17 May 2019). $N_d$ data were obtained from the Centre for Environmental Data Analysis: http://data.ceda.ac.uk/badc/deposited2018/grosvenor_modis_droplet_conc/ (last access: 17 May 2019). DMS measurements from Amsterdam Island were obtained from the World Data Centre of Greenhouse gases: https://gaw.kishou.go.jp (last access: 17 May 2019). DMS measurements from the SOAP campaign can be obtained by contacting Mike Harvey: mike.harvey@niwa.co.nz. DMS measurements from

the SOIREE campaign are available from: Boyd, P. (2009), Cruise data inventory from the R/V Tangaroa 61TG_3052 cruise in the Southern Ocean during 1999 (SOIREE project), Biological and Chemical Oceanography Data Management Office (BCO-DMO), Dataset version 2009-09-17, http://lod.bco-dmo.org/id/dataset/3212 (last access: 17 May 2019). ERA-Interim data were obtained from the European Centre for Medium-Range Weather Forecasts: http://apps.ecmwf.int/datasets/data/interim-full-moda/levtype=sfc/ (last access: 17 May 2019).

*Code and data availability.* Model simulation data are archived at New Zealand eScience Infrastructure (NeSI) www.nesi.org and are avail-
able by contacting the corresponding author.

*Author contributions.* LER implemented model developments, performed model simulations and wrote the manuscript with assistance from all co-authors. SK assisted with experimental design and obtained observational data sets and evaluated the model, together with LER, LB and AS. SH provided the new sea spray aerosol source function tested in the model and performed the binned wind speed-AOD evaluation. MH contributed DMS cruise data and advised on DMS chemistry and aerosols over the Southern Ocean. JPM provided expertise on the
representation of aerosols in HadGEM3. JW provided technical expertise in running model simulations. AJM advised on clouds over the Southern Ocean and the use of remote sensing to evaluate the model. OM and VV contributed expertise in running HadGEM3.

*Competing interests.* The authors declare no competing interests

*Acknowledgements.* We acknowledge the Deep South National Science Challenge for their support of this research (grant C01X1412), and the UK Met Office for the use of the MetUM. We also wish to acknowledge the contribution of New Zealand eScience Infrastructure (NeSI)
high-performance computing facilities to the results of this research. New Zealand's national facilities are provided by NeSI and funded jointly by NeSI's collaborator institutions and through the Ministry of Business, Innovation and Employment's Research Infrastructure programme (www.nesi.org.nz). LER acknowledges China Southern for partial support. We acknowledge the Cape Grim Science Program



for the provision of DMS data from Cape Grim. The Cape Grim Science Program is a collaboration between the Australian Bureau of Meteorology and the CSIRO Australia. We also acknowledge the MISR and MODIS mission scientists and associated NASA personnel for the production of data used in this research effort.



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



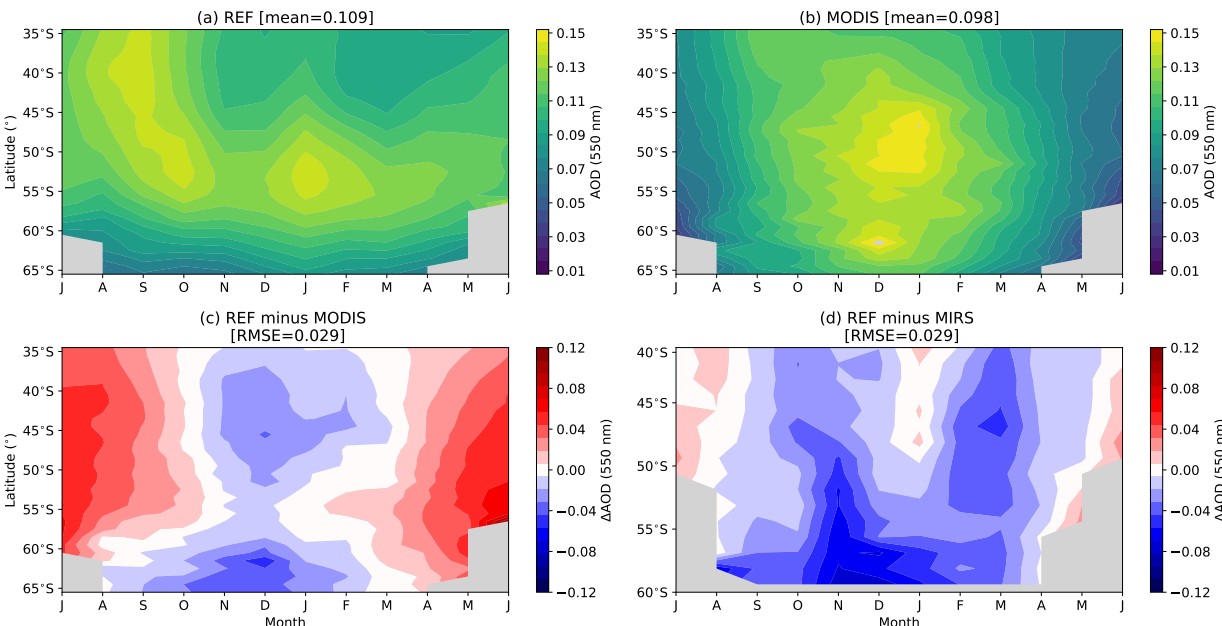

**Figure 1.** (a) Monthly climatological zonal-mean aerosol optical depth (AOD) at 550 nm between 2003–2007 for the REF simulation performed with the modified HadGEM3-GA7.1 model. Daily-mean model data were temporally co-located with daily-mean MODIS satellite data. The grey shaded area indicates where MODIS data are unavailable, and the mean AOD is indicated in the title. (b) As for (a), but showing AOD measured by MODIS-Aqua (the Moderate Resolution Imaging Spectroradiometer) collection 6.1. (c) The difference between HadGEM3-GA7.1-mod and MODIS; i.e. (a) minus (b). The root mean square error (RMSE) is indicated in the title. (d) As for (c), but showing the difference between HadGEM3-GA7.1-mod and MISR (Multi-angle Imaging Spectroradiometer).



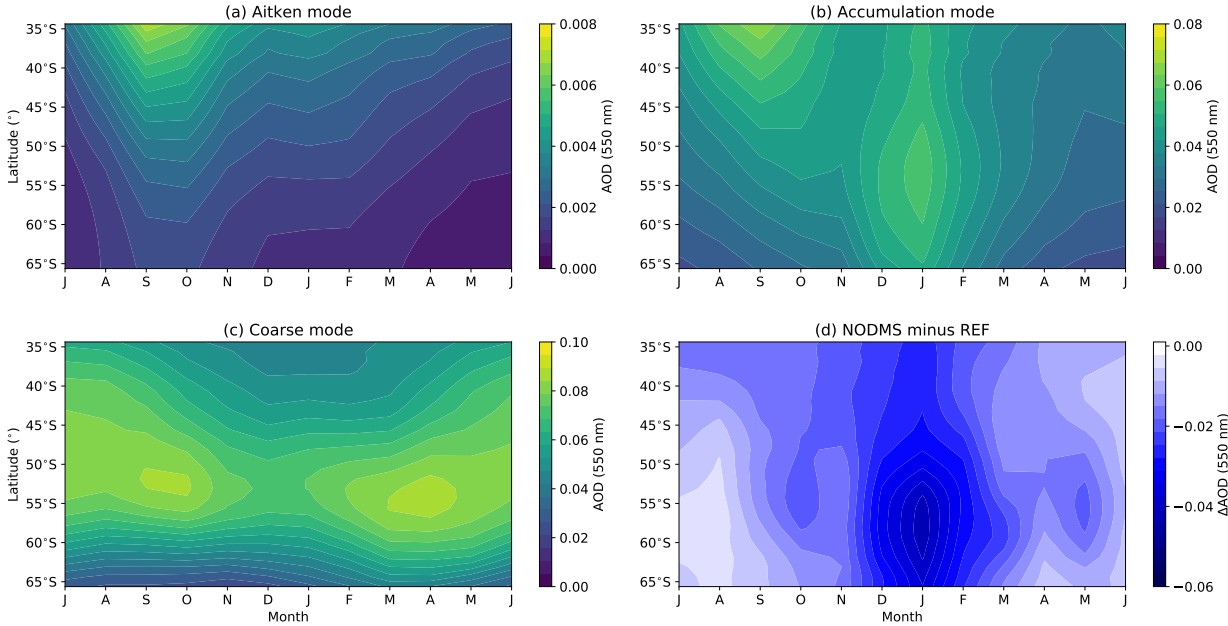

**Figure 2.** (a) Monthly climatological zonal-mean contribution to AOD at 550 nm from soluble Aitken mode AOD in the REF simulation performed with HadGEM3-GA7.1-mod. (b) As for (a), but showing accumulation mode AOD. (c) As for (a), but showing coarse mode AOD. (d) Difference in climatological monthly-mean AOD between the REF simulation and NODMS simulation with surface DMS emissions switched off.



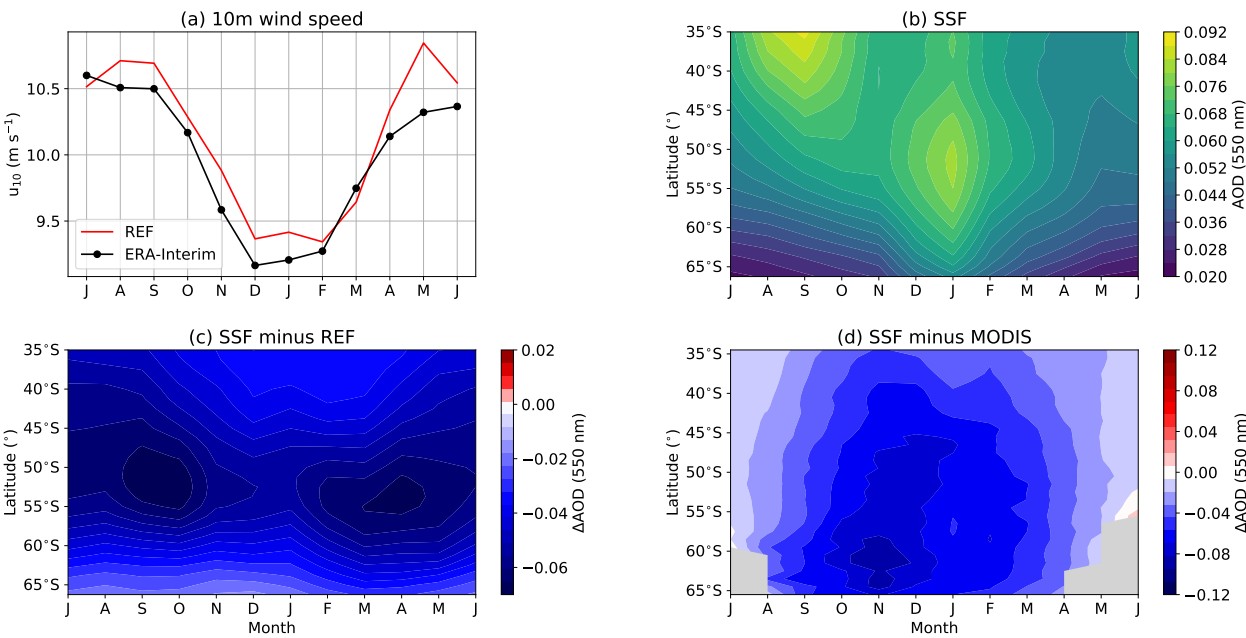

**Figure 3.** (a) Near-surface (10 m) climatological monthly mean wind speed between 40–60°S in the REF simulation and ERA-Interim reanalysis between 2003–2007. (b) Monthly-mean AOD in the SSF sensitivity simulation with the SSA source function changed to that of (Hartery et al.). (c) AOD difference between the REF and SSF simulations. (d) AOD difference between MODIS and the SSF simulation.



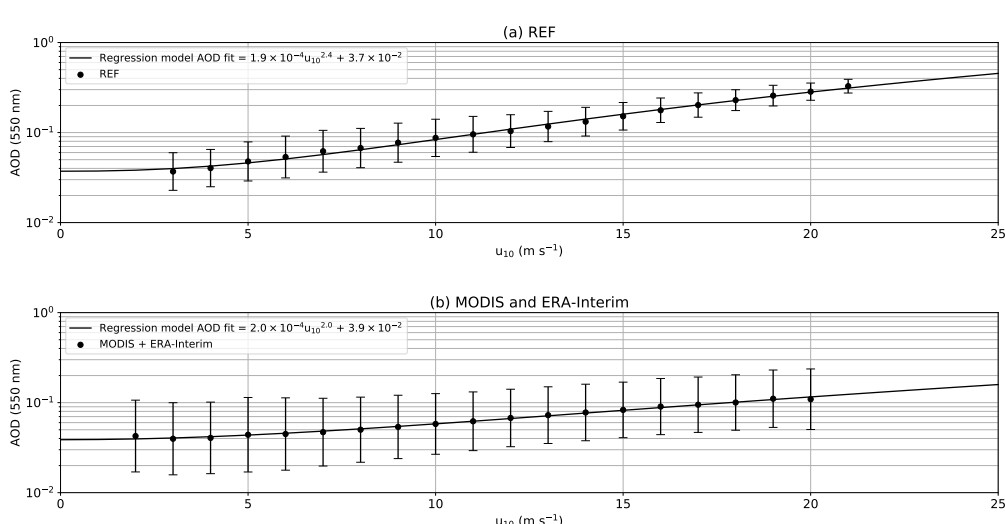

**Figure 4.** (a) Binned AOD at 550 nm vs 10 m wind speed in the REF simulation. Daily averages of AOD were matched to 10 m wind speeds for all ocean grid-cells at latitudes between 40–60°S for July 2003–2007. These values were then sorted into discretised 1 m s$^{-1}$ bins, and the median AOD in each bin was recorded. A least-squares regression was performed on the gridded data and the fit is shown in the legend. (b) As for (a), but showing the 10 m wind speed from ERA-Interim and AOD from MODIS between 2003–2007. Days on which there were less than five observations of AOD in a given $0.5° \times 0.5°$ grid cell were removed from the analysis.

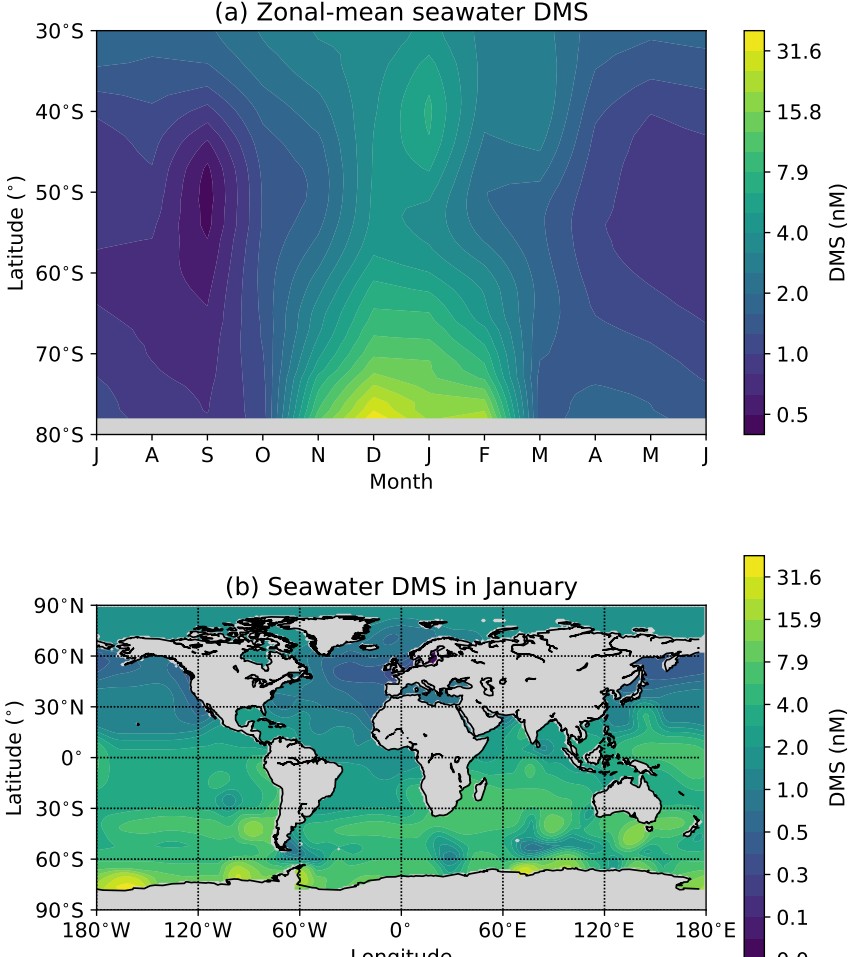

**Figure 5.** Seawater DMS concentrations from the Lana et al. (2011) climatology used as input to HadGEM3-GA7.1-mod shown for (a) the Southern Ocean as a zonal mean; (b) globally for January.

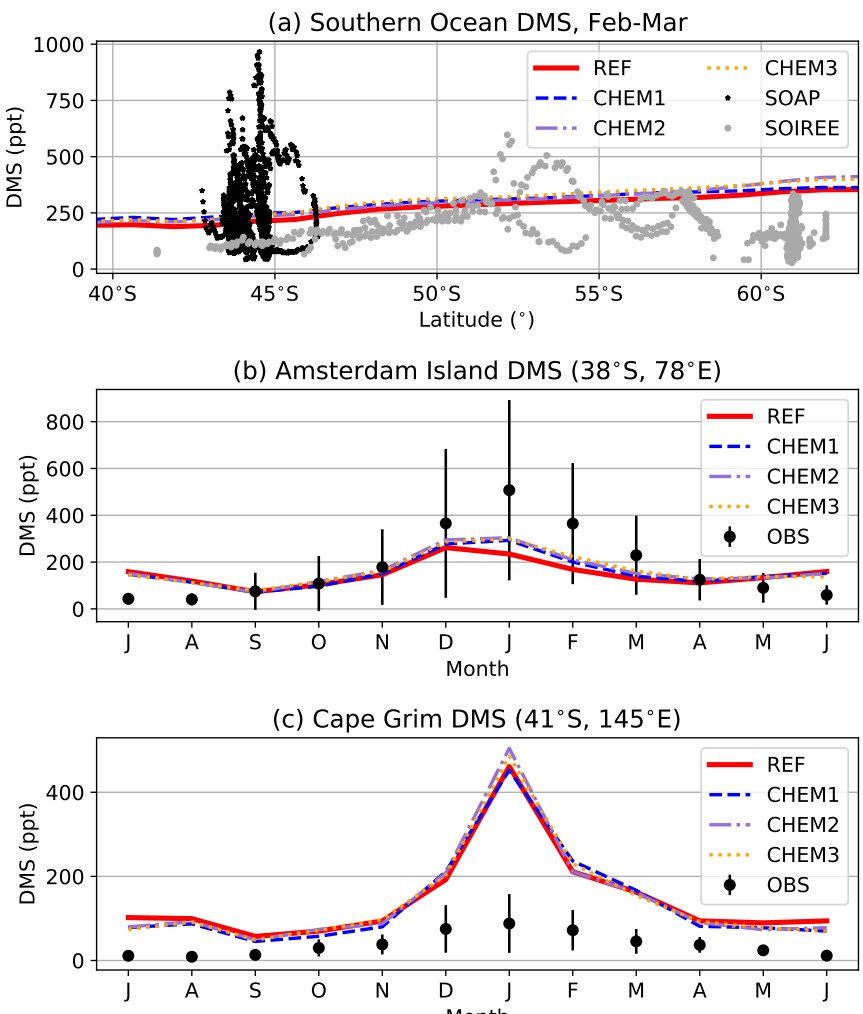

**Figure 6.** (a) DMS measurements obtained from the Surface Ocean Aerosol Production campaign (SOAP) and the Southern Ocean Iron Release Experiment (SOIREE), compared to simulated surface atmospheric DMS concentrations from the REF, CHEM1, CHEM2 and CHEM3 simulations. Model data are averaged over the combined domain of both voyages (172–180°E) and show February-March climatological means calculated over the period 1989-2008. (b) Climatological monthly-mean surface atmospheric DMS measured at Amsterdam Island between 1987–2008 compared to simulated surface atmospheric DMS concentrations at Amsterdam Island in the REF and CHEM simulations. The errorbars show $1\pm\sigma$ on the observed monthly mean. (c) As for (b), but showing DMS at Cape Grim between 1989–1996.



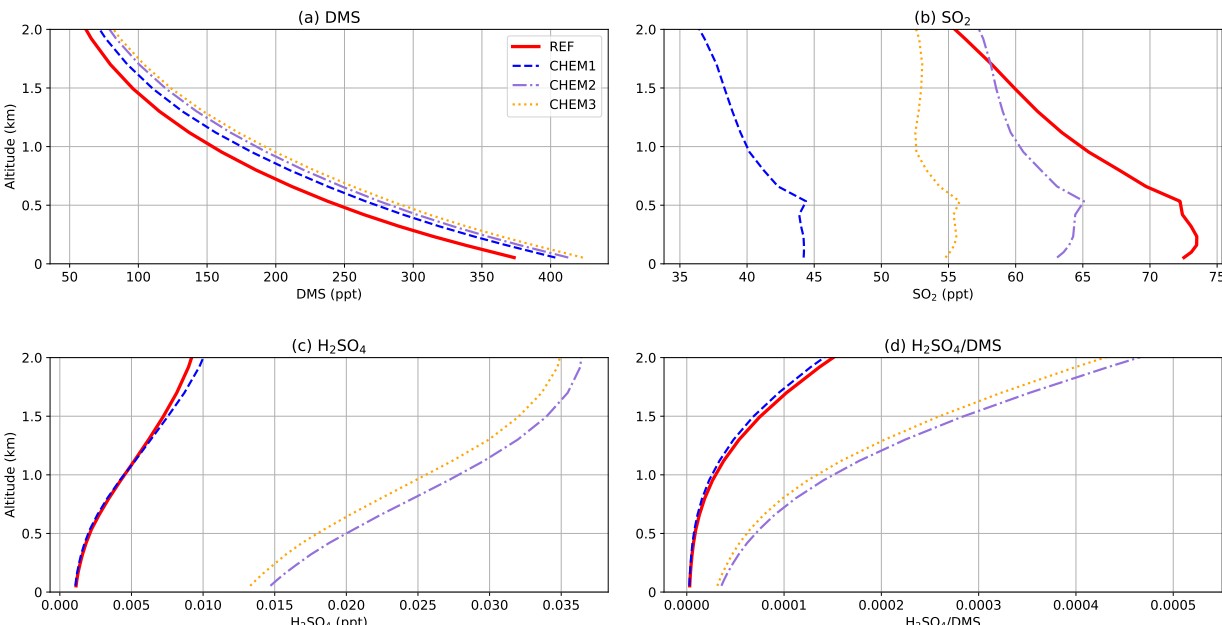

**Figure 7.** Vertical profiles of trace gas mixing ratios between 40–60°S in the lowest 2 km of the atmosphere for the REF and CHEM simulations. All quantities are climatological means for DJF. (a) DMS; (b) $SO_2$; (c) $H_2SO_4$; (d) the ratio of $H_2SO_4/DMS$.



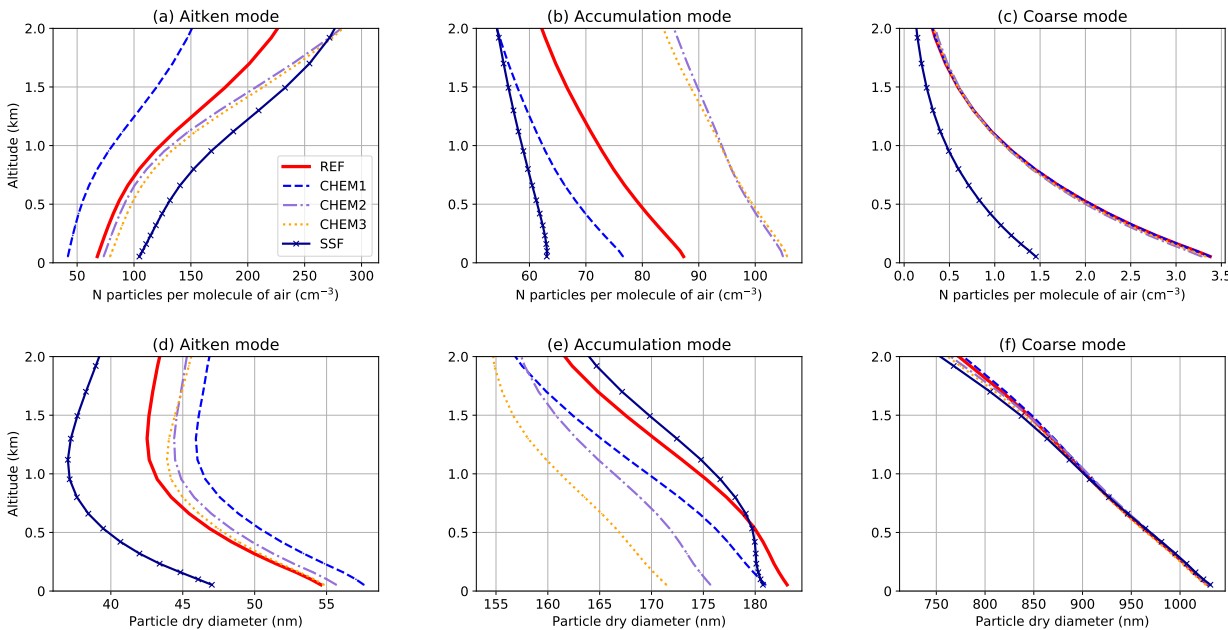

**Figure 8.** Climatological-mean profiles in DJF between 40–60°S in the lowest 2 km of the atmosphere of aerosol mode number concentration (top row) and dry diameter (bottom row) for the REF, CHEM and SSF simulations. (a) and (d) show the soluble Aitken mode; (b) and (e) the accumulation mode; (c) and (f) the coarse mode.





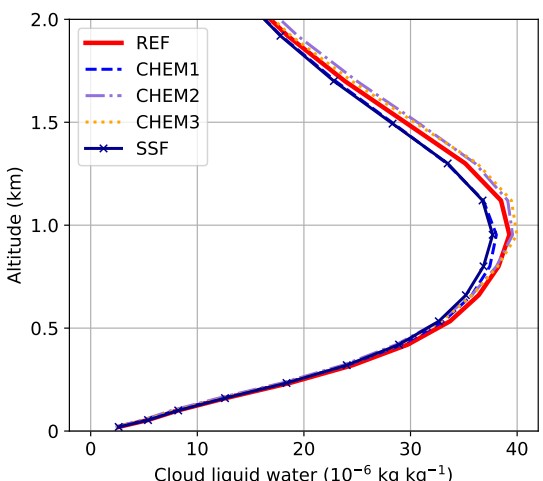

**Figure 9.** The mass fraction of cloud liquid water in DJF between 40–60°S in the lowest 2 km of the atmosphere.





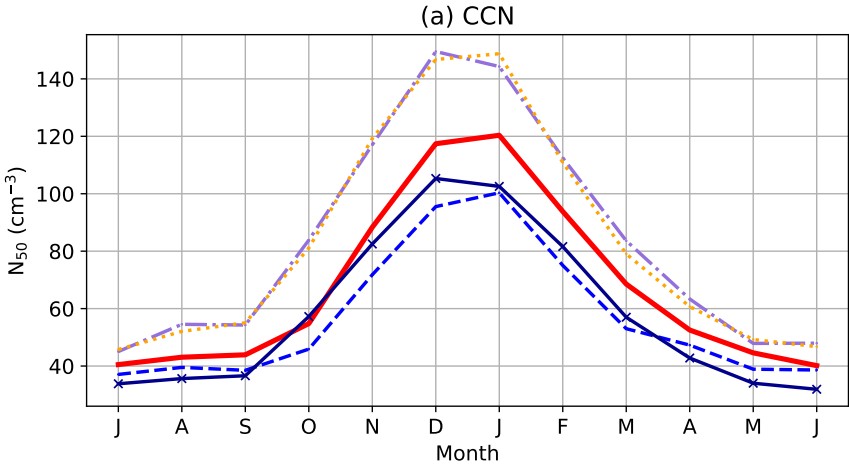

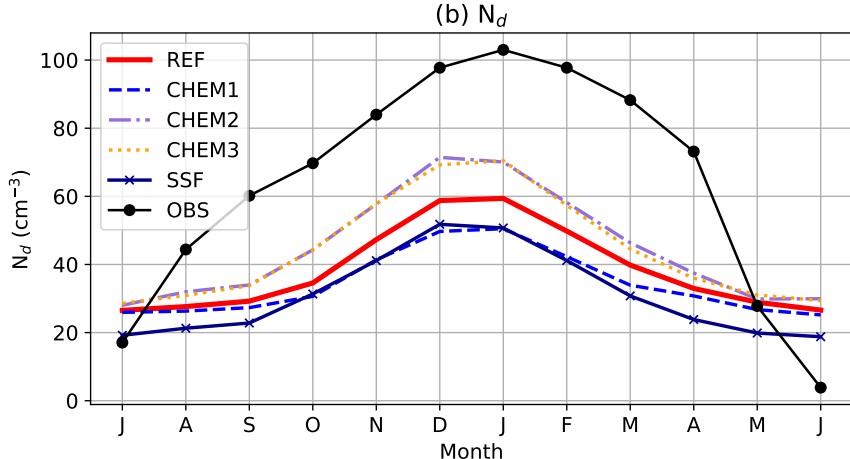

**Figure 10.** (a) Climatological-mean seasonal cycle in the concentration of cloud condensation nuclei at 800 m above the surface between 40–60°S. (b) As for (a), but showing the cloud droplet number concentration at the cloud top. The $N_d$ observations shown are derived from MODIS data (Grosvenor et al., 2018).

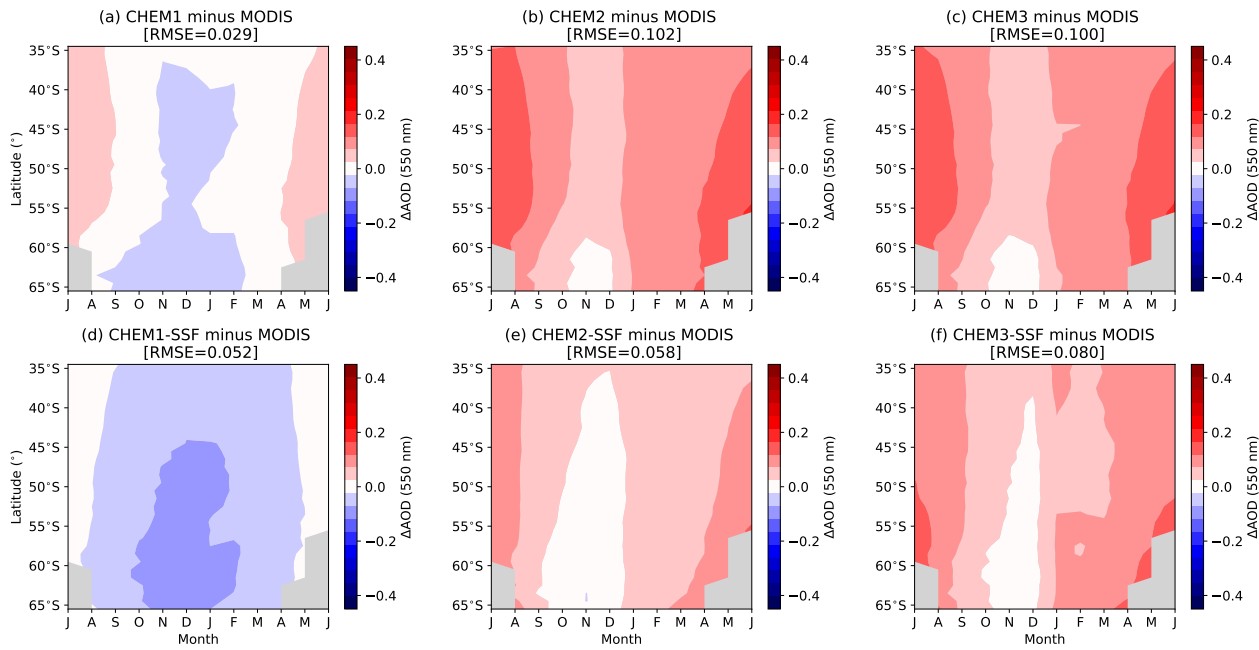

**Figure 11.** The difference in climatological monthly-mean AOD over the Southern Ocean between MODIS and the HadGEM3-GA7.1 sensitivity simulations: (a) CHEM1; (b) CHEM2; (c) CHEM3; (d) CHEM1-SSF; (e) CHEM2-SSF; (f) CHEM3-SSF. The root mean square error (RMSE) is indicated in the title.




**Table 1.** Current StratTrop sulfate chemistry scheme

| Gas-phase reactions | $k_{298}$ [cm$^3$ s$^{-1}$]$^a$ | $-E_a/R$ [K]$^b$ | Reference |
|---|---|---|---|
| DMS + OH $\rightarrow$ SO$_2$ | $1.20\times10^{-11}$ | -260 | Pham et al. (1995) |
| DMS + OH $\rightarrow$ SO$_2$ + MSA | $3.04\times10^{-12}$ | 350 | Pham et al. (1995) |
| DMS + NO$_3$ $\rightarrow$ SO$_2$ | $1.90\times10^{-13}$ | 500 | Pham et al. (1995) |
| DMS + O($^3$P) $\rightarrow$ SO$_2$ | $1.30\times10^{-11}$ | 410 | Sander et al. (2006); Weisenstein et al. (1997) |
| SO$_2$ + OH $\rightarrow$ SO$_3$ + HO$_2$ | See note$^c$ | | Pham et al. (1995) |
| Aqueous-phase reactions | $k_{298}$ [M$^{-1}$ s$^{-1}$] | $-E_a/R$ [K] | Reference |
| HSO$_3^-$ + H$_2$O$_{2(aq)}$ + H$^+$ $\rightarrow$ SO$_4^{2-}$ + 2H$^+$ + H$_2$O$_{(aq)}$ | $7.45\times10^7$ | -4430 | Kreidenweis et al. (2003) |
| HSO$_3^-$ + O$_{3(aq)}$ $\rightarrow$ SO$_4^{2-}$ + H$^+$ + O$_{2(aq)}$ | $3.50\times10^5$ | -5530 | Kreidenweis et al. (2003) |
| SO$_3^{2-}$ + O$_{3(aq)}$ $\rightarrow$ SO$_4^{2-}$ + O$_{2(aq)}$ | $1.50\times10^9$ | -5280 | Kreidenweis et al. (2003) |

$^a$Rate constant at 298 K. $^b$Activation temperature. $^c$ Low-pressure limit: $3.3 \times 10^{-31}(300/T)^{3.3}$ cm$^6$ molecule$^{-2}$ s$^{-1}$; high-pressure limit: $1.5 \times 10^{-12}$ cm$^3$ molecule$^{-1}$ s$^{-1}$





**Table 2.** HadGEM3-GA7.1-mod simulations performed

| Experiment | SSA source function | DMS emission scaling | Gas-phase DMS chemistry | Aqueous-phase sulfate chemistry |
|---|---|---|---|---|
| REF | Gong (2003) | 1 | StratTrop | StratTrop |
| SSF | Hartery et al. | 1 | StratTrop | StratTrop |
| NODMS | Gong (2003) | 0 | StratTrop | StratTrop |
| CHEM1 | Gong (2003) | 1 | StratTrop with DMS+BrO and DMS+Cl | StratTrop |
| CHEM2 | Gong (2003) | 1 | Chen et al. (2018) | StratTrop |
| CHEM3 | Gong (2003) | 1 | Chen et al. (2018) | Chen et al. (2018) |
| CHEM1-SSF | Hartery et al. | 1 | StratTrop with DMS+BrO and DMS+Cl | StratTrop |
| CHEM2-SSF | Hartery et al. | 1 | Chen et al. (2018) | StratTrop |
| CHEM3-SSF | Hartery et al. | 1 | Chen et al. (2018) | Chen et al. (2018) |

In the Gong (2003) SSA source function, SSA generation is dependent on $u_{10}^{3.41}$ while in Hartery et al. it depends on $u_{10}^{2.8}$. See Table 1 for more details of the StratTrop chemistry scheme, and Table 3 for more details of the (Chen et al., 2018) chemistry scheme.



**Table 3.** Reaction schemes tested with the CHEM1, CHEM2 and CHEM3 sensitivity simulations

| Gas-phase reactions | $k_{298}$ [cm$^3$ s$^{-1}$] | $-E_a/R$ [K] | Reference |
|---|---|---|---|
| **CHEM1**: | | | |
| DMS + OH → SO$_2$ + CH$_3$O$_2$ + HCHO | $4.69\times10^{-12}$ | -280 | Burkholder et al. (2015) |
| DMS + OH → 0.6SO$_2$ + 0.4DMSO + CH$_3$O$_2$ | $3.04\times10^{-12}$ | 350 | Pham et al. (1995) |
| DMS + NO$_3$ → SO$_2$ + HNO$_3$ + CH$_3$O$_2$ + HCHO | $1.13\times10^{-12}$ | 530 | Burkholder et al. (2015) |
| DMS + BrO → DMSO + Br | $3.39\times10^{-13}$ | 950 | Burkholder et al. (2015) |
| DMS + O$_3$ → SO$_2$ | $1.00\times10^{-19}$ | 0 | Burkholder et al. (2015); Du et al. (2007) |
| DMS + Cl → 0.5SO$_2$ + 0.5DMSO + 0.5HCl + 0.5ClO | $3.40\times10^{-10}$ | 0 | Burkholder et al. (2015); Barnes et al. (2006) |
| | | | |
| **CHEM2**: As for CHEM1 plus the following reactions: | | | |
| DMSO + OH → 0.95MSIA[a] + 0.05SO$_2$ | $8.94\times10^{-11}$ | 800 | Burkholder et al. (2015); von Glasow and Crutzen (2004) |
| MSIA + OH → 0.9SO$_2$ + 0.1MSA | $9.00\times10^{-11}$ | 0 | Burkholder et al. (2015); Hoffmann et al. (2016) |
| | | | Kukui et al. (2003); Zhu et al. (2006) |
| MSIA + O$_3$ → MSA | $2.00\times10^{-18}$ | 0 | Lucas and Prinn (2002); von Glasow and Crutzen (2004) |
| SO$_2$ + OH → H$_2$SO$_4$ + HO$_2$ | See note[b] | | Burkholder et al. (2015) |

| Aqueous-phase reactions | $k_{298}$ [M$^{-1}$ s$^{-1}$] | $-E_a/R$ [K] | Reference |
|---|---|---|---|
| **CHEM3**: As for CHEM2 plus the following reactions: | | | |
| DMS$_{(aq)}$ + O$_{3(aq)}$ → DMSO$_{(aq)}$ + O$_{2(aq)}$ | $8.61\times10^8$ | -2600 | Gershenzon et al. (2001) |
| MSIA$_{(aq)}$ + O$_{3(aq)}$ → MSA$_{(aq)}$ | $3.50\times10^7$ | 0 | Hoffmann et al. (2016) |
| MSI$^-$ [c] + O$_{3(aq)}$ → MS$^-$ [d] | $2.00\times10^6$ | 0 | Flyunt et al. (2001) |
| HSO$_3^-$ + HOBr$_{(aq)}$ → SO$_4^{2-}$ + 2H$^+$ + Br$^-$ | $3.20\times10^9$ | 0 | Chen et al. (2016, 2017) |
| SO$_3^{2-}$ + HOBr$_{(aq)}$ → SO$_4^{2-}$ + H$^+$ + Br$^-$ | $5.00\times10^9$ | 0 | Troy and Margerum (1991) |
| HSO$_3^-$ + H$_2$O$_{2(aq)}$ + H$^+$ → SO$_4^{2-}$ + 2H$^+$ + H$_2$O$_{(aq)}$ | $7.45\times10^7$ | -4760 | Jacob (1986); Kreidenweis et al. (2003) |
| HSO$_3^-$ + O$_{3(aq)}$ → SO$_4^{2-}$ + H$^+$ + O$_{2(aq)}$ | $3.20\times10^5$ | -4830 | Jacob (1986) |
| SO$_3^{2-}$ + O$_{3(aq)}$ → SO$_4^{2-}$ + O$_{2(aq)}$ | $1.00\times10^9$ | -4030 | Jacob (1986) |

[a] Methanesulfinic acid, CH$_3$SOOH. [b] Low-pressure limit: $3.3\times10^{-31}(300/T)^{4.3}$ cm$^6$ molecule$^{-2}$ s$^{-1}$; high-pressure limit: $1.6\times10^{-12}$ cm$^3$ molecule$^{-1}$ s$^{-1}$ as described by (Chen et al., 2018). [c] CH$_3$SOO$^{-1}$. [d] CH$_3$SO$_2$O$^{-1}$.