# Peer review of "The sensitivity of Southern Ocean aerosols and cloud microphysics to sea spray and sulfate aerosol production in the HadGEM3-GA7.1 chemistry-climate model"

_Atmospheric Chemistry and Physics, 2019_

## Short Comment (SC1) · 21 Aug 2019

Which version of the MISR data was used in the study? A new V23 has been available since June 2018, and this version has greatly improved quality of AOD retrievals over oceans.

---

## Referee Comment (RC1) · Anonymous Referee #1 · 18 Sep 2019

In this study, the authors examine how the aerosol optical depth (AOD), cloud condensation nuclei, and cloud droplet number concentrations calculated in the HadGEM3-GA7.1 chemistry-climate model vary over the Southern Ocean for different sea salt aerosol source functions and DMS oxidation schemes. They compare their results to MODIS and MISR observations of AOD as well as to observations of DMS and satellite-derived cloud droplet number concentrations. The authors find that the model overestimates AOD in winter and underestimates AOD in summer, assigning these differences to an overestimate of sea salt aerosol emission in winter and underestimate

of DMS-derived aerosol during summer. Based on a set of sensitivity studies they recommend a lower sea salt aerosol source function (by Hartery et al., in prep) and the implementation of new DMS chemistry schemes suggested by Chen et al. (2018).

This manuscript is a sensitivity study, as the title indicates. The manuscript is generally well written. The two main concerns I have are: 1) the new source function for sea salt aerosol emissions is based on a study in preparation, and 2) the recommendations of the manuscript are not well justified. I elaborate on these concerns below.

The authors contrast the Gong (2003) source function to one with a windspeed to the power of 2.8 from a study in preparation. Very little information is given from that study, other than it is based on "Analysis of aerosol measurements made on a 2018 Tangaroa research voyage on the Southern Ocean". What type of measurements were these? How was the analysis conducted? How does the new source function compare to observations of SSA mass concentrations available at coastal ground-sites and during other cruises? I have to be a little skeptical of applying the results of one single set of observations in a limited region and season to derive a source function applicable to the entire globe. Furthermore, I am not sure what the policy of ACPD is regarding citations of unpublished results (in prep. or submitted), but many journals do not allow such citations. I am not sure what the course of action is, given that this other study has not undergone peer review and that the current manuscript relies substantially on it.

The authors' main recommendation is "However, given that the chemistry schemes used in the CHEM2 and CHEM3 simulations also show the best agreement with Nd observations, we recommend a combination of the Hartery et al. SSA source function and either the CHEM2 or CHEM3 DMS chemistry schemes for future studies." (page 12). In their recommendation, the authors seem to weight more strongly the comparison to Nd than to AOD. Can they please justify this? Almost no information is provided about the Nd dataset used. The authors reference one paper (Grosvenor et al., 2018) in the figure caption, but that paper discusses multiple retrievals of Nd. What specific

Nd dataset is used, for what year? How accurate are these retrievals (I see not error bars in Figure 10)? How is that Nd dataset compared to the simulations (model sampling)? Also, Figure 10 shows that the Hartery et al. source function (SSF) leads to a decrease in Nd relative to most of the other simulations. I would thus expect that combining CHEM2 or CHEM3 to that SSA scheme would lead to Nd being close to the REF simulation. Hence the comparison between CHEM2-SSF and CHEM3-SSF to observed Nd would not necessarily lead to an improvement. Can the authors add these comparisons to Figure 10 and provide a better justification for their recommendation? As the manuscript stands I am unconvinced that one scheme is better than the other.

Minor comments.

Page 2. Lines 9-10 "...sulfate aerosol are formed from nucleation of sulfur-containing gases". Not all sulfur-containing gases lead to new particle formation. Another pathway is condensation on pre-existing particles.

Page 2. Lines 22-24. In addition to forming $SO_2$, DMS can also form MSA.

Page 5. Lines 13-15 "...emissions of aerosols and their precursors were prescribed based on the year 2000". Is this only for anthropogenic emissions or all emissions? For example, are the SSA emissions based on winds for the year 2000 or do they vary with the actual winds calculated for the specific year?

Figure 1. The latitude range on panel d (60-40S) is different than for all other panels (65-35S), which makes the comparison difficult. Also, the labels on the colorbars do not fall on the discrete colors, making it difficult to relate a given color to a specific value in AOD or AOD difference. The sample problem exists with the colorbars for all the other figures. I suggest that the authors modify the colorbars to make them more readable.

Figure 4. Could the authors combine the 2 panels into one so that the two functions can be compared? Or at least, add the fit from panel a to panel b. Why were the bins

[Figure]

only done for the month of July? The texts mentions that windspeeds and hence SSA are high during all seasons except summer. Does the correlation between windspeed and AOD change for different months? Also, it wasn't clear from the text whether the AOD and windspeeds are averaged longitudinally before doing the binning, or whether individual gridboxes are considered.
* * *

---

## Referee Comment (RC2) · Anonymous Referee #2 · 10 Oct 2019

This study examines model simulations of aerosol optical depth (AOD) and the relative contributions of the Aitken, accumulation, and coarse aerosol size modes to AOD throughout the seasonal cycle. Comparisons are made to MODIS and MISR satellite observations, which indicate that the model is overpredicting the amount of primary sea spray aerosol. The overprediction is attributed to the sea spray source function, and the sensitivity of the model to this source function is tested with a newly-developed empirical model derived from field observations (that is apparently explained in detail in a paper currently in preparation – Hartery et al.). Additional sensitivity simulations

are performed to explore changes in the representation of gas- and aerosol-phase conversion of DMS to sulfate aerosol as described by Chen et al., 2018. Overall, the manuscript is well written, and the topic is relevant to ACP. I share the concerns of the other reviewer that the fundamental sea spray source function employed by this study is based on a paper that has not yet been even submitted, much less in a peer-reviewed form with only a cursory description (and no real validation) provided in this paper. Other than this issue, I recommend the paper for publication.
* * *

---

## Author Comment (AC1) · 28 Oct 2019

The MISR data used are Level 3 (averaged and gridded), version 4 (MISR MIL3MAE v4, https://eosweb.larc.nasa.gov/project/misr/mil3mae_table). This has been clarified in the revised manuscript.
* * *

---

## Author Comment (AC2) · 4 Nov 2019

**Reply to Anonymous Referee #1**

Thanks to the referee for their helpful comments, which we feel have improved the clarity and quality of the manuscript. Point-by-point responses follow below in which the referee's comments are in red, our response is in black, and changes to the text are in blue.

Laura Revell (on behalf of all other co-authors), University of Canterbury, 4 November 2019.

- The authors contrast the Gong (2003) source function to one with a windspeed to the power of 2.8 from a study in preparation. Very little information is given from that study, other than it is based on "Analysis of aerosol measurements made on a 2018 Tangaroa research voyage on the Southern Ocean". What type of measurements were these? How was the analysis conducted? How does the new source function compare to observations of SSA mass concentrations available at coastal ground-sites and during other cruises?

The Hartery *et al.* (2019) sea salt aerosol source function (hereafter H2019) cited in the manuscript is based on a series of *in situ* measurements of the total suspended sea spray concentration within the Southern Ocean boundary layer. The total concentration of sea spray was constrained from the number concentration size spectra measured with a PCASP-100X optical particle counter throughout a voyage from Wellington, New Zealand to the Ross Sea in February-March 2018. (A map of the voyage track is given in Figure 1 of Kuma *et al.* (2019) – the TAN1802 track.)

After the voyage, the Lagrangian particle trajectory model FLEXPART-WRF was used to develop source-receptor relations between the upwind environment and the *in situ* measurements. The source-receptor framework acted as a bridge through which several different formulas for the sea spray source function could be optimized. The newly optimized functions all found that the Gong (2003) parameterization (hereafter G2003) produced too much sea spray at high wind speeds, as described by H2019 and others (e.g. Madry *et al.*, 2011, Jaeglé *et al.*, 2011, Spada *et al.*, 2015).

In our study we used one such function; the newly optimized power-law developed by H2019 (Eq. 5 of our manuscript). We used this parameterization as it takes the same power-law form as G2003 (i.e $a \times u_{10}{}^{b}$) and was thus straightforward to implement into our chemistry-climate model. H2019 show that the two parameterizations differ primarily at high wind speeds; for example when the 10 m scalar wind speed ($u_{10}$) is 4 m s$^{-1}$, H2019 and G2003 both predict the same SSA flux. However when $u_{10}$ = 11 m s$^{-1}$, H2019 predicts a 40% smaller sea salt aerosol flux than G2003, which H2019 demonstrate is more realistic compared to *in situ* measurements.

Finally, H2019 tested predictions for the total number concentrations of sea spray from the optimized functions against airborne data collected on HIAPER (the NSF/NCAR High-performance Instrumented Airborne Platform for Environmental Research) as part of the SOCRATES (Southern Ocean Clouds, Radiation, Aerosol Transport Experimental Study) campaign. The goodness-of-fit between predictions and airborne measurements provided validation that the newly optimized functions accurately constrained the flux of sea spray from the Southern Ocean.

We have included a summary of the above in the methods section of our manuscript, which is given in our response to reviewer 2. In addition, H2019 has been submitted to JGR: Atmospheres.

- I have to be a little skeptical of applying the results of one single set of observations in a limited region and season to derive a source function applicable to the entire globe.

As noted above, the difference between H2019 and the original G2003 parameterisation is primarily at the high wind speeds observed over the Southern Ocean; elsewhere the difference in simulated sea salt aerosol is minimal. However, we have noted in the manuscript the need to evaluate the new model implementations globally at the end of the results and discussion section:

Future work will focus on … evaluating changes to clouds and aerosols outside the Southern Ocean region when these changes are implemented.

- The authors' main recommendation is "However, given that the chemistry schemes used in the CHEM2 and CHEM3 simulations also show the best agreement with $N_d$ observations, we recommend a combination of the Hartery et al. SSA source function and either the CHEM2 or CHEM3 DMS chemistry schemes for future studies." (page 12). In their recommendation, the authors seem to weight more strongly the comparison to $N_d$ than to AOD. Can they please justify this?

One motivation for carrying out this study was to improve the representation of cloud microphysics in the model, which in turn should improve the simulated shortwave radiative bias over the Southern Ocean in the model (e.g. see the original manuscript, page 3 lines 3-7). We agree with the reviewer that our recommendation is unclear and have reworded this section:

Of all the CHEM and CHEM-SSF sensitivity simulations, AOD simulated in the CHEM1 simulation agrees most favorably with MODIS (Fig. 11a), and the root-mean square error between 40-60°S has decreased slightly (from 0.032 to 0.028) following the original REF and MODIS comparison (Fig. 1c). However, the seasonal bias remains. The CHEM1-SSF simulation shows good agreement with MODIS during austral winter but underestimates summertime AOD and $N_d$ (Fig. 10 and 11d). CHEM2-SSF and CHEM3-SSF show the reverse; simulated summertime AOD agrees well with MODIS but wintertime AOD is too high, even with the new SSA source function included. In terms of simulating Southern Ocean AOD accurately, we recommend CHEM1 for future studies. However, given the improvements in $N_d$ in the CHEM2, CHEM3, CHEM2-SSF and CHEM3-SSF simulations relative to observations, the CHEM2 and CHEM3 DMS chemistry schemes allow for a more accurate representation of cloud microphysical properties over the Southern Ocean. Furthermore, the CHEM2 and CHEM3 schemes represent a fundamentally improved representation of DMS chemistry over the default scheme, and improve process-based understanding of sulfate aerosol formation over the Southern Ocean.

- Almost no information is provided about the $N_d$ dataset used. The authors reference one paper (Grosvenor et al., 2018) in the figure caption, but that paper discusses multiple retrievals of $N_d$. What specific $N_d$ dataset is used, for what year? How accurate are these retrievals (I see no error bars in Figure 10)? How is that $N_d$ dataset compared to the simulations (model sampling)?

The $N_d$ dataset used is the "GW14" (Grosvenor and Wood, 2014) data set analysed by Grosvenor et al. (2018), but updated through to 2015. Grosvenor et al. (2018) compare GW14 with the "BR17" (Bennartz and Rausch, 2017) $N_d$ data set. Annual-mean $N_d$ concentrations are shown to be similar in the two data sets over the Southern Ocean (see Fig. 7 of Grosvenor et al. (2018)),

with BR17 reporting slightly higher values. Grosvenor *et al.* (2018) note: "At high latitudes, the BR17 data set shows larger values than those from GW14. This is likely due to the lack of screening for SZA bias in BR17 beyond what is done in the operational MODIS Level 2 cloud product."

Grosvenor *et al.* (2018) $N_d$ data are available for 2003 – 2015. For our study these were compiled into climatological monthly means, and the area-weighted climatological monthly mean is plotted in Fig. 10b. The same procedure was followed for each of the model simulations. Measurement uncertainties are not available with the Grosvenor *et al.* (2018) data set, but we have added the standard deviation on the climatological monthly mean to give an indication of variability (see figure, below).

[Figure]

[Figure]

Figure 10. (a) Climatological-mean seasonal cycle in the concentration of cloud condensation nuclei at 800 m above the surface between 40–60°S. (b) As for (a), but showing the cloud droplet number concentration at the cloud top. The $N_d$ observations shown are derived from MODIS data over the period 2003 – 2015 (Grosvenor *et al.*, 2018). The climatological monthly mean is plotted for all years available of observational data and model data. The errorbars on the observations indicate the standard deviation on the climatological monthly mean.

We have included a summary of the above points in the Methods section of the revised manuscript:

To evaluate $N_d$ in HadGEM3-GA7.1-mod we used the "GW14" data set presented by Grosvenor *et al.* (2018), which is derived from MODIS retrievals. GW14 was originally developed by Grosvenor and Wood (2014) and later extended to 2015. Grosvenor *et al.* (2018) compare the GW14 data set with another MODIS-derived $N_d$ data set compiled by Bennartz and Rausch (2017) and show that the two data sets are similar over the Southern Ocean, with the Bennartz and Rausch (2017) data set reporting slightly higher values. Grosvenor *et al.* (2018) note that this is likely related to a lack of screening for any biases in the solar zenith angle in the Bennartz and Rausch (2017) data set. We compiled GW14 $N_d$ data between 2003 and 2015 into climatological monthly means, and examined the area-weighted mean over the Southern Ocean. The same procedure was followed for each of the model simulations.

- Also, Figure 10 shows that the Hartery et al. source function (SSF) leads to a decrease in $N_d$ relative to most of the other simulations. I would thus expect that combining CHEM2 or CHEM3 to that SSA scheme would lead to $N_d$ being close to the REF simulation. Hence the comparison between CHEM2-SSF and CHEM3-SSF to observed Nd would not necessarily lead to an improvement. Can the authors add these comparisons to Figure 10 and provide a better justification for their recommendation?

In general, the SSF simulations (using H2019) show 5-20% less $N_d$ than their corresponding simulation with G2003. We have added the CHEM1-SSF, CHEM2-SSF and CHEM3-SSF simulations to Figure 10 following the reviewer's suggestion, as shown above. Even when the new SSA parameterisation is included, the CHEM2-SSF and CHEM3-SSF simulations still represent an improvement in $N_d$ over REF relative to observations.

- Page 2. Lines 9-10 ". . .sulfate aerosol are formed from nucleation of sulfur-containing gases". Not all sulfur-containing gases lead to new particle formation. Another pathway is condensation on pre-existing particles.

Thank you. We have noted this in the text:

Secondary aerosols such as sulfate aerosol are formed from nucleation of sulfur-containing gases or condensation on pre-existing particles.

- Page 2. Lines 22-24. In addition to forming SO2, DMS can also form MSA.

This is noted in the text:

When DMS is emitted into the atmosphere, it has a lifetime of 1--2 days and undergoes a series of chemical reactions to form sulfur dioxide ($SO_2$), and is further oxidised to form sulfuric acid and sulfate aerosol. Alternative oxidation pathways result in some DMS sulfur forming methane sulfonic acid (MSA).

- Page 5. Lines 13-15 ". . .emissions of aerosols and their precursors were prescribed based on the year 2000". Is this only for anthropogenic emissions or all emissions? For example, are the SSA emissions based on winds for the year 2000 or do they vary with the actual winds calculated for the specific year?

SSA emissions were calculated based on winds for the year in question (i.e. not only for the year 2000). DMS emissions are calculated from the Lana et al. (2011) seawater climatology and Liss &

Merlivat (1986) DMS sea-air exchange parameterisation (as noted in the original manuscript). All other emissions are for the year 2000, including: $NO_x$, $CH_4$, CO, $SO_2$, isoprene, monoterpenes, $C_2H_6$, $C_3H_8$, formaldehyde, acetaldehyde, acetone, $NH_3$ and biofuel, fossil fuel and biomass burning emissions of black carbon and organic carbon.

We have clarified this in the text:

Emissions of $NO_x$, $CH_4$, CO, $SO_2$, isoprene, monoterpenes, ethane, propane, formaldehyde, acetaldehyde, acetone, $NH_3$, black carbon and organic carbon were prescribed based on the year 2000 (Lamarque *et al.,* 2010).

- Figure 1. The latitude range on panel d (60-40S) is different than for all other panels (65-35S), which makes the comparison difficult. Also, the labels on the colorbars do not fall on the discrete colors, making it difficult to relate a given color to a specific value in AOD or AOD difference. The sample problem exists with the colorbars for all the other figures. I suggest that the authors modify the colorbars to make them more readable.

Following the reviewers suggestion, we have ensured that the latitude range is set to 40-60°S for all contour plots shown. In some cases this has changed the calculated root-mean square errors slightly, and these have been updated in the text where necessary. We have also re-plotted the figures with discrete colorbars, as recommended. The revised figure 1 is shown below:

[Figure]

- Figure 4. Could the authors combine the 2 panels into one so that the two functions can be compared? Or at least, add the fit from panel a to panel b. Why were the bins only done for the month of July? The texts mentions that windspeeds and hence SSA are high during all seasons except summer. Does the correlation between windspeed and AOD change for different months? Also, it wasn't clear from the text whether the AOD and windspeeds are averaged longitudinally before doing the binning, or whether individual gridboxes are considered.

[Figure]

Thanks for this helpful suggestion – we have combined the panels as shown above and it is much clearer to read. July was chosen as a representative wintertime month when wind speeds are high over the Southern Ocean (Fig. 3a) and aerosol predominantly consists of sea salt (Fig. 2d). We have noted both points in the text.

The correlation between wind speed and AOD does indeed change for different months; this has already been established in previous papers (e.g. Madry *et al.,* 2011) and we decided not to show that here. Finally yes, individual gridboxes were considered (i.e. AOD and wind speeds were not averaged longitudinally prior to binning). We have clarified this in the caption:

Figure 4. Binned AOD at 550 nm vs 10 m wind speed. Daily averages of AOD were matched to 10 m wind speeds for all ocean grid-cells at latitudes between 40-60°S for July 2003-2007. These values were then sorted into discretised 1 m s$^{-1}$ bins, and the median AOD in each bin was recorded. Grid cells were considered individually during the binning process (i.e. AOD and wind speeds were not averaged zonally prior to binning). A least-squares regression was performed on the gridded data and the fit is shown for the model and ERA-Interim and MODIS between 2003-2007. Days on which there were less than five MODIS observations of AOD in a given 0.5° x 0.5° grid cell were removed from the analysis.

References

Bennartz, R., & Rausch, J. (2017). Global and regional estimates of warm cloud droplet number concentration based on 13 years of AQUA-MODIS observations. Atmospheric Chemistry and Physics, 17(16), 9815–9836. https://doi.org/10.5194/acp-17-9815-2017.

Gong, S. L.: A parameterization of sea-salt aerosol source function for sub- and super-micron particles, Global Biogeochemical Cycles, 17, https://doi.org/10.1029/2003GB002079, 2003.

Grosvenor, D. P., &Wood, R. (2014). The effect of solar zenith angle on MODIS cloud optical and microphysical retrievals within marine liquid water clouds. Atmospheric Chemistry and Physics, 14, 7291–7321. https://doi.org/10.5194/acp-14-7291-2014.

Hartery, S., Toohey, D., Revell, L., Sellegri, K., Kuma, P., Harvey, M., and McDonald, A.: Constraining the surface flux of sea spray aerosol from the Southern Ocean, Journal of Geophysical Research: Atmospheres, submitted, 2019.

Jaeglé, L., Quinn, P. K., Bates, T. S., Alexander, B., and Lin, J.-T.: Global distribution of sea salt aerosols: new constraints from in situ and remote sensing observations, Atmospheric Chemistry and Physics, 11, 3137–3157, https://doi.org/10.5194/acp-11-3137-2011, 2011.

Kuma, P., McDonald, A. J., Morgenstern, O., Alexander, S. P., Cassano, J. J., Garrett, S., Halla, J., Hartery, S., Harvey, M. J., Parsons, S., Plank, G., Varma, V., and Williams, J.: Evaluation of Southern Ocean cloud in the HadGEM3 general circulation model and MERRA-2 reanalysis using ship-based observations, Atmospheric Chemistry and Physics Discussions, 2019, 1–37, https://doi.org/10.5194/acp-2019-201, 2019.

Madry, W. L., Toon, O. B., and O'Dowd, C. D.: Modeled optical thickness of sea-salt aerosol, Journal of Geophysical Research: Atmospheres, 116, https://doi.org/10.1029/2010jd014691, 2011.

Spada, M., Jorba, O., Pérez García-Pando, C., Janjic, Z., and Baldasano, J. M.: On the evaluation of global sea-salt aerosol models at coastal/orographic sites, Atmospheric Environment, 101, 41–48, https://doi.org/10.1016/j.atmosenv.2014.11.019, 2015.

---

## Author Comment (AC3) · 4 Nov 2019

**Reply to Anonymous Referee #2**

Thanks to the referee for their positive comments. Our response is below, in which the referee's comments are in red, our response is in black, and changes to the text are in blue.

Laura Revell (on behalf of all other co-authors), University of Canterbury, 4 November 2019.

This study examines model simulations of aerosol optical depth (AOD) and the relative contributions of the Aitken, accumulation, and coarse aerosol size modes to AOD throughout the seasonal cycle. Comparisons are made to MODIS and MISR satellite observations, which indicate that the model is overpredicting the amount of primary sea spray aerosol. The overprediction is attributed to the sea spray source function, and the sensitivity of the model to this source function is tested with a newly-developed empirical model derived from field observations (that is apparently explained in detail in a paper currently in preparation – Hartery et al.). Additional sensitivity simulations are performed to explore changes in the representation of gas- and aerosol-phase conversion of DMS to sulfate aerosol as described by Chen et al., 2018. Overall, the manuscript is well written, and the topic is relevant to ACP. I share the concerns of the other reviewer that the fundamental sea spray source function employed by this study is based on a paper that has not yet been even submitted, much less in a peer-reviewed form with only a cursory description (and no real validation) provided in this paper. Other than this issue, I recommend the paper for publication.

The paper describing the new sea spray source function that we test has now been submitted to the Journal of Geophysical Research: Atmospheres (Hartery *et al.,* 2019). In our response to Referee #1 we documented the types of measurements made by Hartery *et al.* (2019), how their analysis was conducted, and how they validated their function. We also included information regarding this in the methods section of the revised manuscript:

The Hartery *et al.* (2019) SSA source function is based on a series of *in situ* measurements of the total suspended sea spray concentration within the Southern Ocean boundary layer. The total concentration of sea spray was constrained from the number concentration size spectra measured with a PCASP-100X optical particle counter during a voyage from Wellington, New Zealand, to the Ross Sea in February-March 2018.

After the voyage, the Lagrangian particle trajectory model FLEXPART-WRF was used to develop source-receptor relations between the upwind environment and the *in situ* measurements. The source-receptor framework acted as a bridge through which several different formulas for the sea spray source function could be optimised. The newly optimised functions all found that the Gong (2003) parametrisation produced too much sea spray at high wind speeds, as described by Hartery *et al.* (2019) and previous studies including Madry *et al.* (2011), Jaeglé *et al.* (2011) and Spada *et al.* (2015).

One of the newly optimised parametrisations developed by Hartery *et al.* (2019) took a power-law form (i.e. Eq. 5), similar to the Gong (2003) parametrisation (Eq. 2). We selected this parametrisation to test as it was straightforward to implement in HadGEM3-GA7.1. Hartery *et al.* (2019) show that the two power-law parametrisations differ primarily at high wind speeds, which are commonly observed over the Southern Ocean. For example when $u_{10}$ = 4 m s$^{-1}$, both parametrisations predict the same SSA flux. However when $u_{10}$ = 11 m s$^{-1}$, the Hartery *et al.* (2019) SSA parametrisation predicts a SSA flux which is 40% smaller than that predicted by Gong (2003).

Hartery *et al.* (2019) validated their newly optimised parametrisations by comparing predicted SSA concentrations against airborne data collected on HIAPER (the NSF/NCAR High-performance Instrumented Airborne Platform for Environmental Research) as part of the SOCRATES (Southern Ocean Clouds, Radiation, Aerosol Transport Experimental Study) campaign. The goodness-of-fit between predictions and airborne measurements validated the use of the new parametrisations over the Southern Ocean.

References

Gong, S. L.: A parameterization of sea-salt aerosol source function for sub- and super-micron particles, Global Biogeochemical Cycles, 17, https://doi.org/10.1029/2003GB002079, 2003.

Hartery, S., Toohey, D., Revell, L., Sellegri, K., Kuma, P., Harvey, M., and McDonald, A.: Constraining the surface flux of sea spray aerosol from the Southern Ocean, Journal of Geophysical Research: Atmospheres, submitted, 2019.

Jaeglé, L., Quinn, P. K., Bates, T. S., Alexander, B., and Lin, J.-T.: Global distribution of sea salt aerosols: new constraints from in situ and remote sensing observations, Atmospheric Chemistry and Physics, 11, 3137–3157, https://doi.org/10.5194/acp-11-3137-2011, 2011.

Madry, W. L., Toon, O. B., and O'Dowd, C. D.: Modeled optical thickness of sea-salt aerosol, Journal of Geophysical Research: Atmospheres, 116, https://doi.org/10.1029/2010jd014691, 2011.

Spada, M., Jorba, O., Pérez García-Pando, C., Janjic, Z., and Baldasano, J. M.: On the evaluation of global sea-salt aerosol models at coastal/orographic sites, Atmospheric Environment, 101, 41–48, https://doi.org/10.1016/j.atmosenv.2014.11.019, 2015.